# TIME-AWARE WORLD MODEL:
# ADAPTIVE LEARNING OF TASK DYNAMICS

## ABSTRACT

In this work, we introduce Time-Aware World Model, a model-based approach designed to explicitly incorporate the temporal dynamics of environments. By conditioning on the time step size, $\Delta t$, and training over a diverse range of $\Delta t$ values – rather than relying on a fixed time step size – our model enables learning of both high- and low-frequency task dynamics in real-world control problems. Inspired by the information-theoretic principle that the optimal sampling rate varies depending on the underlying dynamics of different physical systems, our time-aware model enhances both performance and learning efficiency. Empirical evaluations demonstrate that our model consistently outperforms baseline approaches across different observation rates in various control tasks, using the same number of training samples and iterations. We will release our source code on GitHub once the final review decisions are made.

## 1 INTRODUCTION

Deep reinforcement learning (DRL), one of the most popular learning paradigms, offers a broad range of potential applications, including robotics (Wu et al., 2023; Koh et al., 2021; Johannink et al., 2019), autonomous vehicles (Kiran et al., 2021; Guan et al., 2024), and challenging control tasks where classical approaches fail to deliver satisfactory performance (Prasad et al., 2017; Nhu et al., 2023; Yang et al., 2015). In addition to model-free RL approaches, where an agent learns to directly map current observation $s_t$ to action $a_t$ (Williams & Peng, 1989; Schulman et al., 2015; 2017; Haarnoja et al., 2018), there has been increasing attention on model-based RL (MBRL), which involves training a model $\mathcal{M}$ to capture the underlying dynamics of a given task (Sutton, 1990; Deisenroth & Rasmussen, 2011; Parmas et al., 2018; Kaiser et al., 2019; Janner et al., 2019), and planning the actions by leveraging the trained model $\mathcal{M}$, which we refer to as a *world model*. These model-based approaches have gained significant traction over model-free methods due to their superior sample efficiency and improved generalization capabilities (Ha & Schmidhuber, 2018; Hafner et al., 2019; 2020; 2023; Hansen et al., 2022; 2024).

Despite the impressive performance of world models in various RL tasks, a crucial factor in handling dynamical systems – namely the time step size, $\Delta t$ – has been overlooked in existing work, to the best of our knowledge. Specifically, the dynamics model $\mathcal{D} : s_t, a_t \rightarrow s_{t+1}$, a key component of the world model, models the state transitions. Conventional methods typically train $\mathcal{D}$ using experience tuples $(o_t, a_t, o_{t+1}, r_t)$ collected through interactions with the environment at a fixed time step. However, this practice presents two primary limitations (Thodoroff et al., 2022):

1. **Temporal resolution overfitting**: In current training pipelines, the default simulation time step $\Delta t$ is very small ($\Delta t = 0.0025$ or frequency $f = 400Hz$). While smaller $\Delta t$ values are beneficial for stabilizing simulations and preventing system aliasing, world models trained exclusively on small $\Delta t$ often suffer significant performance degradation when deployed in real-world scenarios with lower observation rates (e.g., $\Delta t = 0.02$ or $f = 50Hz$). This discrepancy presents a major challenge in extending the applicability of world models beyond simulated environments.

2. **Inaccurate system dynamics**: Training the dynamics model $\mathcal{M}$ on a fixed time step $\Delta t$ can lead to overfitting, as the model may not capture the true underlying dynamics of the task without conditioning on $\Delta t$. This can result in inaccurate state transitions and diminished generalization capabilities.

Although some recent works have considered the effect of temporal gap on the state transitions of the model such as (Shaj et al., 2023; Lutter et al., 2021), these works focus on improving the model accuracy over long-horizon predictions, which consists of multiple predictions on a fixed-$\Delta t$ world model. As a result, they trained world models for a single, fixed, discretized time step $\Delta t$ without explicitly considering the impacts of $\Delta t$'s on the state transitions. To overcome these limitations, we primarily focus on addressing the following question: *How can we efficiently train the world model $\mathcal{M}$ to accurately capture the underlying task dynamics across varying time step sizes, while maintaining computational efficiency?*

In this work, we propose a time-aware world model $\mathcal{M}^T$ to address the aforementioned question. Unlike the previous world models $\mathcal{M}$, our model conditions estimation of the next state and reward on $\Delta t$, as they depend on the temporal gap between the current and next state. We formulate $\mathcal{M}^T$ by modifying the world model of TD-MPC2 (Hansen et al., 2024) using 4-th order Runge-Kutta (RK4) method (Butcher, 1987) to enforce certain dynamical properties as explained in Section 4.1. Additionally, we modify the value model to take $\Delta t$ as an extra input. We train these models using various values of $\Delta t$, which are log-uniformly sampled from a predefined interval.

Although one might anticipate that $\mathcal{M}^T$ would require more training samples than $\mathcal{M}$ due to the inclusion of the additional parameter $\Delta t$, this is not necessarily the case. According to the Nyquist-Shannon sampling theorem (Shannon, 1949; Jerri, 1977), a signal with the highest frequency $f$ can be reconstructed by sampling it at a minimal frequency just slightly greater than $2f$. Therefore, if the observation rate is much higher than $2f$, the surplus data become redundant – i.e., do not substantially contribute to training the world model. In general, a physical environment consists of multiple dynamical systems of varying frequencies (Section 3.2.1). Therefore, using a mixture of time step sizes during the training process, we effectively expose the model to different sampling frequencies, allowing such sub-systems to be learned more efficiently (Section 3.2.3).

Inspired by the Nyquist-Shannon sampling theorem (Section 3.2.2), *we empirically prove that our time-aware model, with a mix of time steps in observation data sampled during training, achieves much better performance on learning the world model with different time steps at inference time, after the same amount of training time as the baseline model*. We demonstrate the results on diverse control problems in Meta-World (Yu et al., 2020) environments. Our contributions can be summarized as follows:

1. We highlight the importance of a time-aware world model that conditions the dynamic modeling on the time step size $\Delta t$, a critical quantity within a dynamical system. Specifically, by taking into account the temporal information $\Delta t$, the dynamic model learns to capture the underlying task dynamics across a spectrum of time step sizes. This approach is particularly suitable for real-world control problems, where the observation rate can be varying and/or much lower than the default simulation rate.

2. Motivated by the Nyquist-Shannon sampling theorem, we propose a mixture-of-time-step training framework for the time-aware world model without increasing the number of training steps. This approach introduces a novel perspective on training a world model by considering varying sampling rates, which can guide future works in designing more efficient training strategies.

3. Empirically, we show that our time-aware world model can effectively solve control tasks under varying observation rates (e.g.: $\Delta t = 0.03, or f = 33.3Hz$) without the need for additional data. This capability helps to narrow the gap between simulation environments and real-world control applications.

## 2 RELATED WORK

Although model-free RL algorithms have gained popularity due to their impressive performance, their inherent lack of sample efficiency limits their applicability to a broader range of real-world problems (Sutton et al., 1999; Williams & Peng, 1989; Barto et al., 1983; Schulman et al., 2015; 2017). To address this limitation, several approaches have been proposed, including the integration of analytical gradients with lower variance into the policy learning process (Suh et al., 2022; Xu et al., 2022; Son et al., 2024). In contrast, MBRL methods mitigates this issue at a fundamental level by training a dynamics model that *simulates* the real environment, allowing agents to *predict*

future states as well as outcomes of their actions (Deisenroth et al., 2013). Since prediction is not constrained by real-world sampling, MBRL methods are inherently more sample-efficient. These methods differ primarily in (1) how they define the world model $\mathcal{M}$, and (2) how they leverage $\mathcal{M}$ for training or planning. Historically, Gaussian processes (GP)(Deisenroth & Rasmussen, 2011; Parmas et al., 2018) have been widely used, though recently neural networks are preferred over them(Ha & Schmidhuber, 2018; Hafner et al., 2019; 2020; 2023; Hansen et al., 2022; 2024) due to their superior representational power. Among these, the Dreamer model and its variants (Hafner et al., 2019; 2020; 2023) train agents directly using the learned model, whereas Model Predictive Control (MPC)-based methods (Hansen et al., 2022; 2024) rely on planning algorithms to determine actions. One closely related work, Multi Time Scale World Models (MTS3) (Shaj et al., 2023), explicitly considers the temporal gaps as the motivation in learning task dynamics. MTS3, denoted as $\mathcal{M}$, is trained to capture state transitions over different prediction horizons $H$. However, their approach differs from ours significantly: while they account for multiple temporal resolutions, the model is still trained using a single fixed time step size $\Delta t$, and MTS3, at least at the current stage, can handle only a limited number of time scales (in the paper, MTS2 learn fast and slow dynamics at 2 timescales: $\Delta t$ and $H\Delta t$). In contrast, our method incorporates a continuous-valued $\Delta t$ directly into the model, allowing it to predict state transitions across a range of temporal gaps in just one prediction step. This enables our model to predict transitions over large temporal gaps (e.g.: $\Delta t = 30$ms) that would typically require 12-20 smaller prediction steps in models trained using a fixed, small $\Delta t$. Our approach builds upon the state-of-the-art world model TD-MPC2 framework (Hansen et al., 2024) but differs in that our world model is explicitly conditioned on the time step size $\Delta t$ and is trained on a mixture of time step sizes, unlike most prior approaches, which assume a fixed time step size.

## 3 BACKGROUND AND MOTIVATIONS

### 3.1 MODEL BASED REINFORCEMENT LEARNING

We define the control problem as a Markov decision process (MDP), defined by a tuple $(S, A, P, r, \gamma)$, where $S$ is a set of states, $A$ is a set of actions, $P : S \times A \times S \rightarrow \mathbb{R}$ is the ground truth (stochastic) state transition model, $r : S \times A \rightarrow \mathbb{R}$ is the ground truth reward model, and $\gamma$ is the discount factor. Note that we describe $P$ and $r$ as the *ground truth* model, as we will learn and use the state transition and reward model within our world model later, which is described in Section 4.1. The goal of RL is to find a policy that maximizes the expected sum of discounted reward along a state-action trajectory $\tau = \{s_0, a_0, ..., s_{H-1}, a_{H-1}, s_H\}$, where $H$ is the trajectory length. Formally, we need to obtain a policy or planner $\pi$ that maximize:

$$\eta(\pi) = \mathbb{E}_{s_0, a_0, ... \sim \pi} \left[ \sum_{t=0}^{\infty} \gamma^t r(s_t, a_t) \right]. \tag{1}$$

In the context of MBRL, for each task, we train a world dynamic model, which includes the state transition function $d_\phi : S \times A \rightarrow S$ and reward function $r_\phi : S \times A \rightarrow \mathbb{R}$. The trained world model can be employed in several ways to derive the policy, such as using a planner like TD-MPC2. We present our model formulation and training pipeline used to train the time-aware world model in Section 4.1.2.

### 3.2 THEORETICAL MOTIVATIONS

Before introducing our model, we provide theoretical motivations to explain the sample efficiency of using a mixture of time step sizes during the training process of the time-aware world model.

### 3.2.1 MULTI-SCALE DYNMICAL SYSTEMS

In many control problems, the environment dynamics can be decomposed into multiple sub-dynamical systems (or subsystems), each of which can evolve at different temporal scales (Weinan, 2011). In other words, such subsystems can be characterized by different functions with different highest frequencies. Formally, consider a general dynamic system: $\dot{x} = f(x, u, t)$, where $x, u, t$ are

the state, control input, and time, respectively. Using the Euler method, the next state $x'$ is:

$$x' = x + f(x, u, t) \cdot \Delta t \tag{2}$$

where $\Delta t$ is the time step size. Based on the concept of multi-scale dynamical systems, the above state transition can be re-written as:

$$x' = x + \sum_i f_i(x, u, t) \cdot \Delta t \tag{3}$$

where $f_i(x, u, t)$ represents the dynamic of each sub-system $i$. Each sub-system $f_i(\cdot)$ may evolve at different temporal scales, thus each of them can have its own highest frequency $f_i^{max}$.

### 3.2.2 NYQUIST-SHANNON SAMPLING THEOREM

The Nyquist-Shannon sampling theorem states that a signal must be sampled at a rate at least twice the highest frequency present to avoid losing information (i.e., to prevent aliasing): $f_{sample} > 2f_{max}$ (Shannon, 1949), where $f_{sample}$ and $f_{max}$ are the observation rate and the highest frequency of the environment dynamics in MBRL context, respectively. If the observation rate is too low such that $f_{sample} < 2f_{max}$, we lose important dynamics details caused by the large temporal gap $\Delta t$. This loss causes high-frequency components to be folded back into lower frequencies, resulting in inaccurate dynamics learned by the world model (Zeng et al., 2024). Although higher $f_{sample}$ allows for more accurate reconstruction of the environment dynamics, if $f_{sample}$ is excessively high, oversampling introduces redundant data, increasing sample complexity and reducing learning efficiency. Therefore, finding the right observation rate $f_{sample}$ is crucial to balance modeling accuracy and sample efficiency.

### 3.2.3 SIMULTANEOUSLY TRAINING ON MULTIPLE TEMPORAL RESOLUTIONS

Motivated by the need to more effectively sample the observations according to task dynamics of different sub-systems running at different frequencies while preserving the model accuracy, we propose to simultaneously train the world model on multiple temporal resolutions by varying the observation rates $f_{sample}$ (by varying $\Delta t$) during the training process. Specifically, as shown in Equation 3, the underlying task dynamics can consist of several sub-systems $f_i(\cdot)$ operating on different maximum frequencies $f_i^{max}$. According to the Nyquist-Shannon sampling theorem, each sub-system can be most efficiently learned with a different $f_{sample}$. As a result, by randomly varying $f_{sample}$ during the training process, we avoid under-sampling high-frequency components and over-sampling low-frequency components, thereby training sub-systems $f_i(\cdot)$ efficiently. As a result, our time-aware world model can learn the underlying task dynamics at different temporal resolutions without requiring additional data, as shown in Section 5. In the next section, we present our time-aware model and training framework in detail.

## 4 METHODOLOGY

In this section, we present our time-aware model formulation and training pipeline designed to effectively learn a world model that can perform well across various observation rates. Our work focuses on developing a novel time-aware training method that can be seamlessly integrated into any existing world model architecture, enhancing the model's robustness to observation rate variations during inference. We adopt TD-MPC2 (Hansen et al., 2024) as the baseline, adapting its architecture to train time-aware world models for different control

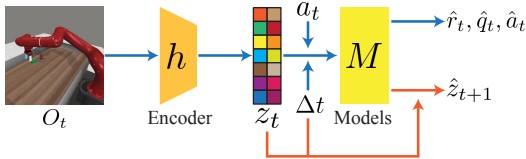

Figure 1: Overall framework of our world model. An encoder $h$ encodes the given observation $s_t$ into a latent vector $z_t$, which is then fed into various models with action $a_t$ and time step size $\Delta t$ to estimate values for action planning.

tasks. First, we provide a high-level overview of the baseline TD-MPC2 and its key architectural components for the sake of completeness. The overall framework is shown in Figure 1.

## 4.1 MODEL ARCHITECTURE

### 4.1.1 STANDARD WORLD MODEL

TD-MPC2 is an MBRL algorithm that learns to capture the underlying task dynamics in the latent space, or "implicit world model". Specifically, unlike other reconstruction-based world model architectures such as World Model (Ha & Schmidhuber, 2018) and DreamerV3 (Hafner et al., 2023), TD-MPC2 does not include a decoder module that maps latent space back to raw observation space. The primary motivation for the exclusion of the decoder module is that it is computationally inefficient to recover the latent space to high-dimensional raw observation space, which can have many elements irrelevant to the control tasks. Once the latent dynamics are learned, TD-MPC2 can roll out different latent predictions for local trajectory optimization with planning algorithms as in MPC (Hansen et al., 2022; 2024). The baseline TD-MPC2 model consists of five key components:

| | | |
|---|---|---|
| Encoder model: | $z_t = h(s_t)$ | *Encode raw observations to latent vectors* |
| Latent dynamic model: | $\hat{z}_{t+1} = D(z_t, a_t)$ | *Latent forward dynamics / transition* |
| Reward model: | $\hat{r}_t = R(z_t, a_t)$ | *Predict immediate reward of state-action pair* |
| Terminal value model: | $\hat{q}_t = Q(z_t, a_t)$ | *Predict state-action value pair* |
| Policy prior: | $\hat{a}_t = p(z_t)$ | *Output optimal action $a_t$ from current latent state* |

where $s_t$ is the raw observation in the current time step, $z_t$ is the latent encoding of the raw observation, $\hat{z}_{t+1}$ is the predicted next latent state, $\hat{r}_t$ is the predicted immediate reward, $\hat{q}_t$ is the estimated q-value of current state-action pair, and $\hat{a}_t$ is the action sampled from the policy for current state. At inference time, the Model Predictive Path Integral (MPPI) planner, a Model Predictive Control (MPC) algorithm, is used for planning and action generation. All five component models of TD-MPC2 are represented by multilayer perceptrons (MLPs).

To train TD-MPC2, a sample buffer $\mathcal{B}$ records trajectories of $\{(s_t, a_t, o_{t+1}, r_t)_{0:H}\}$ from the task environment after each training episode, where $H$ is the episode length. At the end of each training episode, model parameters are updated using data randomly sampled from $\mathcal{B}$. The encoder model $h$, dynamic model $D$, reward model $R$, and terminal value model $Q$ are trained simultaneously through self-supervised consistency loss, supervised reward loss, and supervised temporal-difference terminal value loss, which is described in detail in Hansen et al. (2024). The agent then continues interacting with the task environment and collects additional data for $\mathcal{B}$ to train the model.

### 4.1.2 TIME-AWARE WORLD MODEL

One notable limitation of TD-MPC2 and other state-of-the-art world models is that they ignore the effects of observation rate $\Delta t$ on the accuracy and performance of the world model at inference time. To incorporate time awareness into the model, we propose to condition all components of the world model on $\Delta t$, which can be described as follows:

| | | |
|---|---|---|
| Encoder model: | $z_t = h(s_t)$ | · *Not conditioned on $\Delta t$* |
| Latent dynamic model: | $\hat{z}_{t+1} = z_t + d(z_t, a_t, \Delta t) \cdot \tau(\Delta t)$ | · *State transition* |
| | where where $\tau(x) = max(0, log(x) + 5)$ | *via time-stepping* |
| Reward model: | $\hat{r}_t = R(z_t, a_t, \Delta t)$ | · *Conditioned on $\Delta t$* |
| Terminal value model: | $\hat{q}_t = Q(z_t, a_t, \Delta t)$ | · *Conditioned on $\Delta t$* |
| Policy prior: | $\hat{a}_t = p(z_t, \Delta t)$ | · *Conditioned on $\Delta t$* |

Our proposed time-aware model formulation is architectural-agnostic and can be straightforwardly adapted into any state-of-the-art world model. Since the observation encoder only encodes raw observation to latent space, it does not model the underlying dynamics and thus is not conditioned on $\Delta t$. For all other models that depend on the underlying dynamics, we condition them on the time step size by explicitly using $\Delta t$ as part of the model input.

While the baseline latent dynamic model $D$ directly maps current state-action pair $(z_t, a_t)$ to the next latent state $z_{t+1}$ in an end-to-end manner, we reformulate the dynamic model following the Euler integration method: $\hat{z}_{t+1} = D(z_t, a_t, \Delta t) = z_t + d(z_t, a_t, \Delta t) \cdot \tau(\Delta t)$, where $d(\cdot)$ is represented by an MLP. The motivation for using the Euler method is to naturally enforce an intrinsic condition of a dynamical system: $z_{t+1}|_{\Delta t=0} = z_t \ \forall z_t, a_t$. Specifically, using our latent dynamic model formulation, $\hat{z}_{t+1}|_{\Delta t=0} = z_t + d(z_t, a_t, 0) \cdot \tau(0) \Rightarrow \hat{z}_{t+1}|_{\Delta t=0} = z_t \ \forall z_t, a_t$.

---

**Algorithm 1** Time-Aware World Model Training

---

1: Initialize task environment $\mathcal{E}$, time-aware world model $\mathcal{M}^T$
2: Set experience buffer $\mathcal{B} \leftarrow \emptyset$
3: **repeat**
4:    **for** each *episode* **do**
5:       Set $\Delta t \sim$ Log-Uniform$(0.001, 0.05)$       $\triangleright$ The default $\Delta t$ is $0.0025$
6:       Set $step \leftarrow 0$
7:       **while** $step <$ Horizon $H$ **do**
8:          $a_t \leftarrow$ TDMPC2.act$(s_t, \Delta t)$
9:          Execute $a_t$ in environment $\mathcal{E}$, get back $(s_{t+1}, r_t)$ after $\Delta t$ seconds
10:        Add transition $(s_t, a_t, s_{t+1}, r_t, \Delta t)$ to buffer $\mathcal{B}$
11:        Update world model $\mathcal{M}^T$ using $\{(s_t, a_t, r_t, s_{t+1}, \Delta t)_{1:B}\} \sim \mathcal{B}$
12:        $step \leftarrow step + 1$
13:       **end while**
14:    **end for**
15: **until** reach $N$ training steps
16: **return** $\mathcal{M}^T$

---

Therefore, instead of directly learning transition function $D : (z_t, a_t) \rightarrow z_{t+1}$, we learn the latent state derivative function $d$ (or gradient), which is also conditioned on $\Delta t$ for higher-order derivatives, and then integrate by one step using the Euler method.

**State Transition.** Instead of integrating each time step by $\Delta t$, our dynamic model integrates the dynamics by $\tau(\Delta t) = max(0, log(\Delta t) + 5)$. The reason is that $\Delta t$ can span a wide range, with minimum (e.g: $\Delta t = 10^{-3}$ s) and maximum values (e.g: $\Delta t = 0.5 \times 10^{-1}$ s) differing by orders of magnitude. This wide variation introduced significant numerical challenges for the learning process, as the latent vectors numerically change minimally between time steps, requiring the dynamic model $d(z_t, a_t, \Delta t)$ to scale appropriately across different $\Delta t$. In fact, emperically the model would fail to converge in some tasks when integration with $\Delta t$ used. To address this issue, we assume that the latent state space can be learned to evolve with respect to the logarithm of the time step, $\tau(\Delta t)$, which squashes the values of $\Delta t$ into smaller range, resolving the numerical issue and facilitating efficient model learning. In our experiments, instead of using the Euler method, with the same motivation, we adopt the 4th-order Runge-Kutta (RK4) integration method for our dynamic model. By using RK4, we impose additional learning constraints on the consistency between intermediate latent states, which encourages the model to learn the dynamics across different temporal rates.

## 4.2 Training Pipeline Using a Mixture of Time Resolutions

Algorithm 1 summarizes our framework, which trains a time-aware world model by varying the observation rate during the training process, encouraging the model to learn underlying dynamics at different temporal resolutions. A world model learns the spatial-temporal representation of the environment by sampling observations of these multiple dynamical systems within the environment, at some particular instances in time and space. According to the Nyquist-Shannon sampling theorem, we must sample the "signal" or the information that represents the given World Model at no less than $1/2f$, where $f$ represents the highest frequency of the information in a band-limited signal. However, since there is no systematic methodology to determine the highest frequency of each dynamical system in the environment, *we sample observations from the environment at different temporal rates to more effectively learn the underlying dynamics*.

As shown in Algorithm 1, at the beginning of each episode, we log-uniformly sample $\Delta t$ from an interval and then set the observation rate of that episode as $1/\Delta t$. Log-uniform sampling facilitates sampling observation at higher frequencies (or smaller $\Delta t$) early in the training, which helps stabilize the learning process. If the model is trained with dominantly low observation rates early in the training, it can fail to capture important properties of dynamics and thus harm the learning process. In fact, Figure 4 shows that when the baseline is trained on only a low observation rate ($\Delta t \geq 0.01$s, or 10ms), the models fail on all tasks for all observation rates. Therefore, *mixing different time*

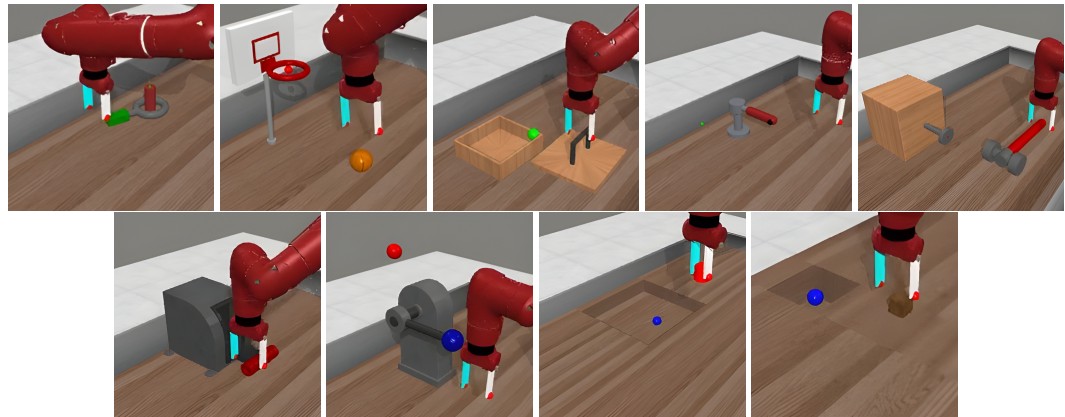

Figure 2: Visualizations of the Meta-World control tasks. From left to right, top row: *Assembly, Basketball, Box Close, Faucet Open, Hammer*; bottom row: *Handle Pull, Lever Pull, Pick Out Of Hole, Sweep Into*. The default time step size of all environments is $\Delta t = 0.0025$s.

*step sizes during the training process not only enhances the model's performance across different observation rates but also helps it to converge efficiently.*

## 5 EXPERIMENTS

We designed our experiments to address three main questions: **(1)** Using the same planner, does the time-aware world model perform comparably to the baseline model for the default observation rate ($\Delta t$) without experiencing performance degradation under lower observation rate? **(2)** For which observation rate (at which $\Delta t$) does the time-aware world model outperform or underperform compared to the baseline? **(3)** Does the time-aware world model require more training samples than the baseline to train?

**Training Environments.** To evaluate the performance and learning efficiency of our proposed time-aware world model, we conducted our experiments on several control tasks within the Meta-World simulation environments, which have a default time step size of $\Delta t = 0.0025$s (2.5ms). (Yu et al., 2021). We adaptively adjusted the observation rate, or the time step $\Delta t$ between observations, during the training process to train the time-aware model. All other settings of the environments are kept as default. We used 9 diverse tasks with different goals and motion characteristics: (1) *Assembly*, (2) *Basketball*, (3) *Box Close*, (4) *Faucet Open*, (5) *Hammer*, (6) *Handle Pull*, (7) *Lever Pull*, (8) *Pick Out Of Hole*, and (9) *Sweep Into*. The task visualizations are depicted in Figure 2. Following Hansen et al. (2024), we use the success rate (%) as the primary metric to measure the performance of the time-aware and baseline models on Meta-World control tasks.

**Training Setup.** Since our time-aware model architecture is based on TD-MPC2 architecture, we kept the *same* TD-MPC2's default training hyperparameters, including model size, learning rate, horizon, etc., to train our model. As shown in Algorithm 1, we used a mixture of different observation rates during the training process by randomly varying the time step size $\Delta t$. Since there is currently no systematic methodology to extract the highest frequency of each control task dynamics, we cannot systematically determine the lowest possible observation rate for the agent to complete the task. Therefore, we empirically set the upper bound of $\Delta t$ to be 0.05s (or 50ms), which is 20× the default time step. We can set the lower bound of $\Delta t$ to be any reasonably small value, which is 0.0001s in our experiments. All of our time-aware models are trained with 1.5M training steps, *fewer* than 2M training steps of the baseline models. Each model completed training in 40-45 hours using a single NVIDIA RTX4000 GPU, 16GB RAM, and 32 CPU cores.

**Performance comparisons across different $\Delta t$ / observation rates.** To evaluate the performance of the time-aware world model on different inference-time observation rates, for each control task, we test the trained models in task environments with different $\Delta t$ settings. From Figure 3, we observe that our time-aware model outperforms the baseline (trained on fixed default $\Delta t = 2.5$ms)

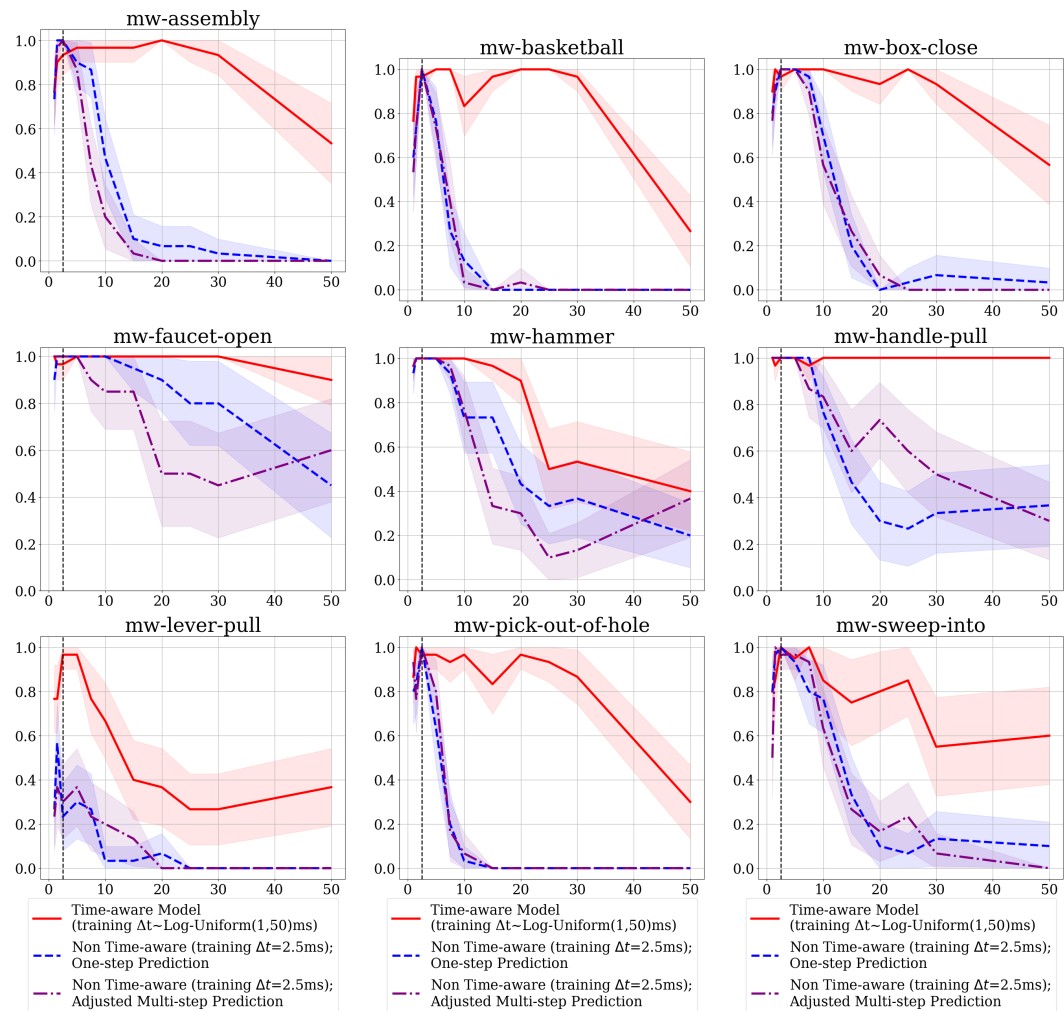

Figure 3: **Task Success Rate under varying observation/action rates.** Average Episode Success Rate as a function of evaluation time steps (unit: millisecond). The dashed lines represent the default time step sizes ($\Delta t = 2.5$ms). Our time-aware model outperforms the baseline (trained on fixed default $\Delta t$) on most evaluation time steps across all tasks while requiring *less training steps* (ours trained with 1.5M steps in RED vs baseline trained with 2M steps in BLUE) with *same hyperparameters*. For fair evaluation, we also adjusted the evaluations for the baselines by repeatedly applying the baselines $\Delta t_{eval}/\Delta t_{train}$ times every time step, which is shown in PURPLE curves. The mean and $95\%$ confidence intervals are plotted over 3 seeds, each with 10 evaluation episodes.

on most evaluation time steps across all tasks, while ours requiring *less* training steps trained with 1.5M steps vs baseline trained with 2M steps – using the *same hyperparameters*. Therefore, our model can effectively learn both underlying *fast* and *slow* dynamics efficiency without increasing sample complexity.

**Effects of using Mixtures of Time Step Sizes.** To demonstrate the effectiveness of training the world model with multiple temporal resolutions $\Delta t$, we compare our model to baseline models trained only on various fixed $\Delta t$, which are different from the default $\Delta t = 2.5$ms. Figure 4 shows that our time-aware model outperforms all the baselines across different control tasks. Most notably, when trained only on low observation rates (e.g: $\Delta t \geq 10$ms), the baseline models cannot converge and fail all the tasks at all observation rates. Therefore, by training the model with a mixture of time step sizes, our time-aware world model effectively outperforms the baseline trained only on a single time step size regardless of the fixed $\Delta t$ value. These results suggest that a dynamical world model can consist of many different dynamical systems, each of which can be described as a time-

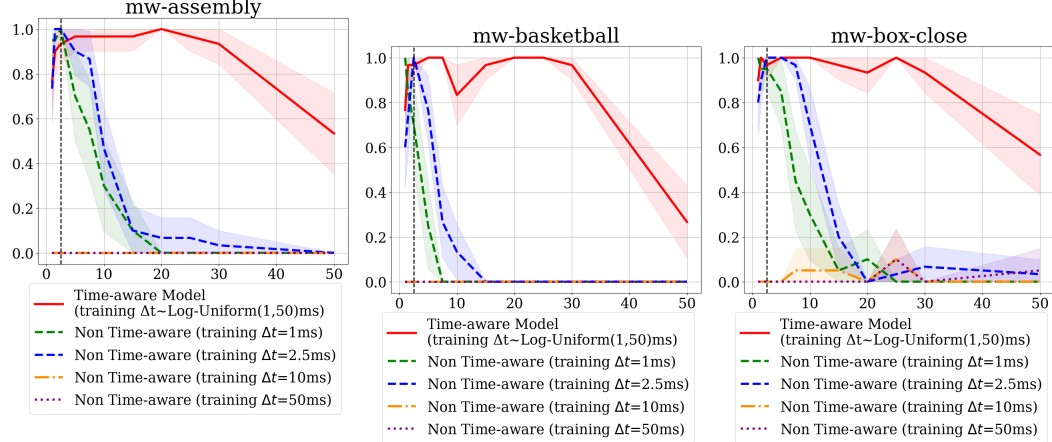

Figure 4: **Task Success Rate under varying observation/action rates.** Average Episode Success Rate as a function of evaluation time steps (unit: millisecond). *Our time-aware model (RED CURVE) outperforms all the baseline models trained on different fixed time step sizes. When trained with only low observation rates ($\Delta t \geq 10$ms, or 0.01s), the non-time-aware models fail on all tasks.* The dashed lines represent the default time step sizes ($\Delta t = 2.5$ms). The mean and $95\%$ confidence intervals are plotted over 3 seeds, each with 10 evaluation episodes.

dependent, parameterized function in space. *Each of such functions can have a different highest frequency, thus by varying the observation rate (or varying $\Delta t$), we allow the world model to more effectively learn such underlying sub-systems.*

**Convergence Rate on Various Inference Timestep Sizes.** To investigate the data efficiency of the time-aware model, we investigate the episode reward/success rate curves across different tasks between our time-aware model and the baseline model, which is trained only on a fixed, default $\Delta t = 2.5$ms. We compare the reward curves of time-aware models and baselines under different evaluation $\Delta t$s across different tasks. From Figure 5, we observe that despite having to adaptively learn more mappings between state transitions under various time step sizes, our time-aware model converges at least as fast as the baseline on most tasks when evaluated on $\Delta t_{default} = 2.5ms$ (the exact $\Delta t$ for which the baseline was specifically trained on, so the best performance of the baseline is expected at this exact $\Delta t$). When evaluated on inference $\Delta t$ different from $\Delta t_{default}$, our time-aware model clearly outperforms the non time-aware baselines by large margins. *These results demonstrate that while having to learn the underlying task dynamics across different temporal resolutions, our time-aware world model does not require additional training steps or samples to converge to a sufficiently accurate model that can effectively solve control tasks at different observation rates.*

## 6 CONCLUSION

In this paper, we introduce a novel *time-aware world model* that can adaptively learn the task dynamics. we show that by explicitly incorporating the time step size $\Delta t$ into the world model and training the model on a mixture of temporal resolutions helps the model to perform robustly under different observation rates on various control tasks without increasing the sample complexity. Empirical results show that our model outperforms all the baseline models, which are trained only on a fixed time step size $\Delta t$ for different training $\Delta t$ values. We hope that the insights and results presented in this paper offer a new perspective on world model training, thereby contributing to the community a new, efficient, yet simple training method to train world models.

**Limitations and Future Work.** One limitation of our work is that we have yet to develop a reliable, systematic methodology to compute analytically the highest frequency of the underlying task dynamics, thus the upper bound for the training time step size $\Delta t_{max}$ is determined empirically. This limitation requires a search of $\Delta t_{max}$ for a new set of environments (e.g.: autonomous driving) to minimize the amount of observations required to train a given task. An interesting avenue for future

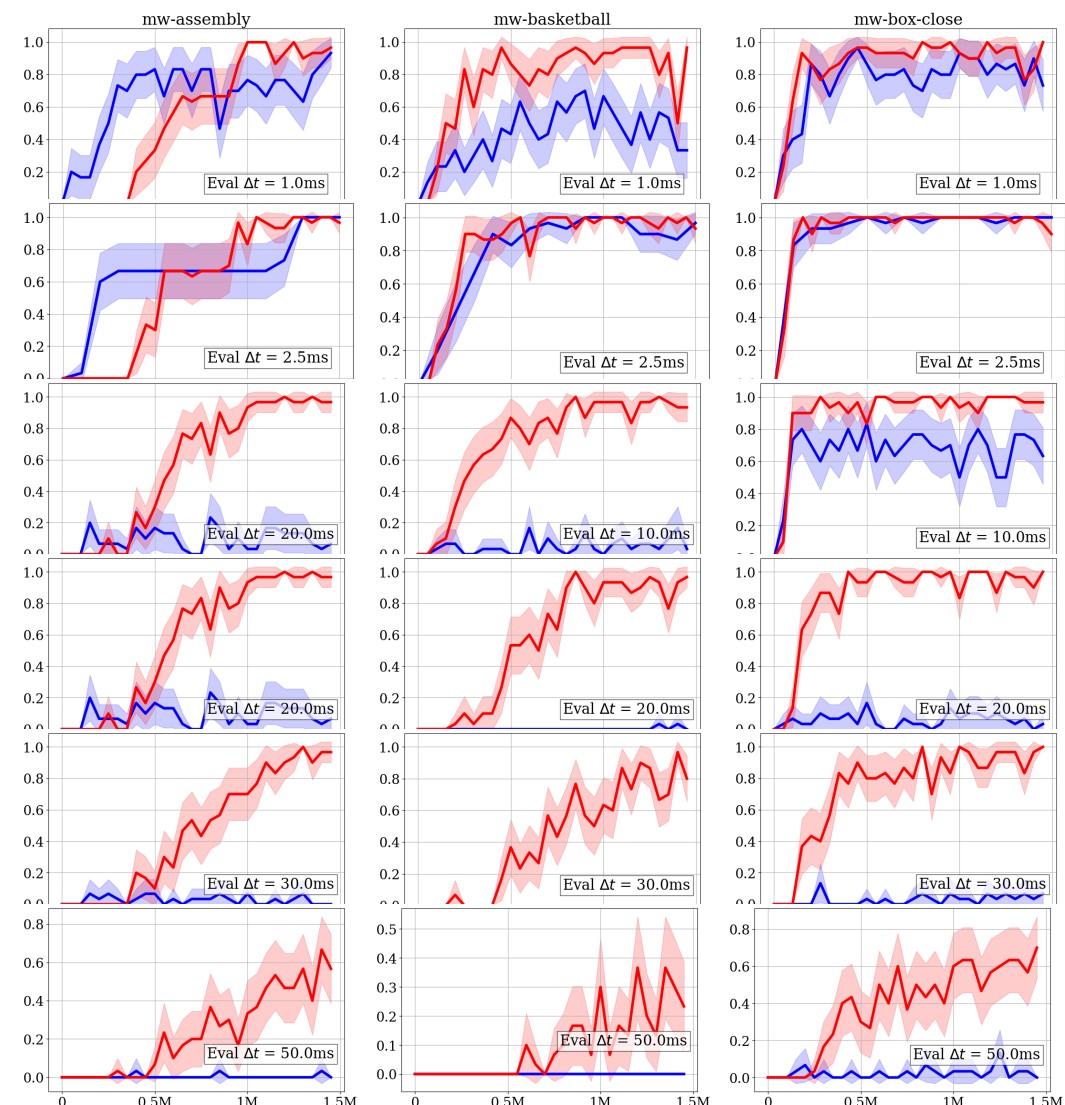

Figure 5: **Success Rate Curve under different evaluation time step sizes.** At each training step, the models are evaluated on various inference $\Delta t$s. Despite having to learn the dynamics under varying time step sizes, our time-aware model (in RED) still converges faster when evaluated on $\Delta t_{default} = 2.5$ms all tasks compared to the baseline trained only on $\Delta t_{default} = 2.5$ms (in BLUE). On large $\Delta t$'s, the time-aware model significantly outperforms the baseline with the same number of training steps while the baselines fail to converge. The mean and 95% confidence intervals success rate are plotted over 3 seeds, each with 10 evaluation episodes.

work is to develop an automatic methodology to find the highest frequency the underlying task dynamics operates on, thereby ensuring the lowest sampling frequency (or highest $\Delta t_{max}$) to train the time-aware world model. Furthermore, we will also adapt our current deterministic dynamic model into a probabilistic one, which is closer to state transitions in the real world, especially at larger temporal gaps.

## REPRODUCIBILITY STATEMENT

To support open science and ensure reproducibility, we will release the code source on GitHub once the paper has been accepted. The implementation details and hyperparameters will be listed and will be set as default in the code source.

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

# Appendix

## A DESCRIPTIONS OF 4TH-ORDER RUNGE-KUTTA INTEGRATION.

In this section, we extend our description of the 4th-order Runge-Kutta (RK4) integration mentioned in Section 4.1.2. The detailed RK4 integration is as follows:

$$
\begin{aligned}
k_1 &= d(z_t, a_t, \Delta t) \\
\hat{z}_1 &= z_t + d\left(z_t, a_t, \Delta t/2\right) \cdot \tau\left(\Delta t/2\right) \\
k_2 &= d\left(\hat{z}_1, a_t, \Delta t\right) \\
\hat{z}_2 &= z_t + d\left(z_1, a_t, \Delta t/2\right) \cdot \tau\left(\Delta t/2\right) \\
k_3 &= d\left(\hat{z}_2, a_t, \Delta t\right) \\
\hat{z}_3 &= z_t + d\left(z_2, a_t, \Delta t\right) \cdot \tau(\Delta t) \\
k_4 &= d\left(\hat{z}_3, a_t, \Delta t\right) \\
\hat{z}_{t+1} &= z_t + \frac{1}{6}(k_1 + 2k_2 + 2k_3 + k4) \cdot \tau(\Delta t)
\end{aligned}
\tag{4}
$$

Consistent with the notations in Section 4.1.2, $z_t, a_t$ denotes the latent state-action pairs at time $t$, $d(\cdot)$ denotes our dynamic model parameterized by a neural network, and $\hat{z}_i$ ($i \in 1, 2, 3$) are the intermediate middle points. The final prediction of next latent state under time step size $\Delta t$ is $\hat{z}_{t+1}$.

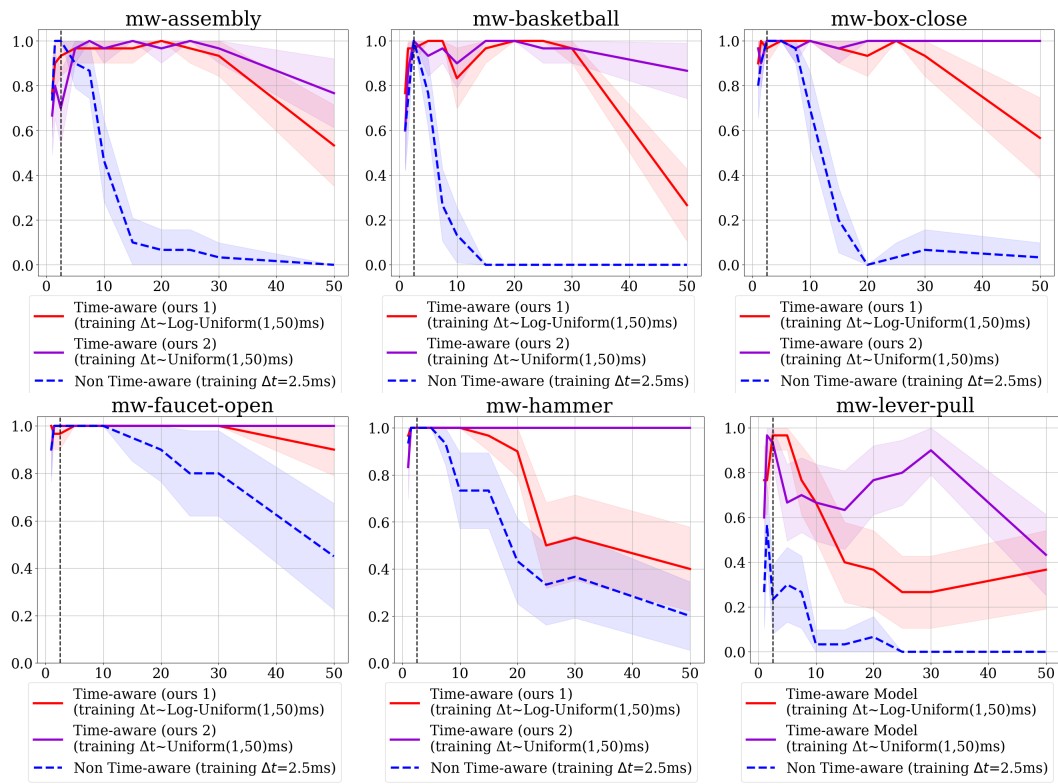

Figure 6: **Ablation Study on $\Delta t$ Sampling Strategy**. Average Episode Success Rate as a function of evaluation time steps (unit: millisecond). The dashed lines represent the default time step sizes ($\Delta t = 2.5$ms). The mean and $95\%$ confidence intervals are plotted over 3 seeds, each with 10 evaluation episodes. The baseline models in BLUE are trained with 2M steps. The time-aware models are trained with 2 different $\Delta t$ sampling strategies: log-uniform sampling in RED and uniform sampling in PURPLE.

## B   ABLATION STUDY OF $\Delta t$ SAMPLING STRATEGIES.

In this section, we conduct an ablation study on the impact of $\Delta t$ sampling strategy during the training process on the performance and efficiency of our time-aware model. Specifically, we trained the same time-aware model architecture under the same range of training $\Delta t's$ but with different sampling strategies: **(1)** Log-Uniform(1,50)ms and **(2)** Uniform(1,50)ms. The performance of time-aware models trained with different $\Delta t's$ sampling strategies (ours) and the non time-aware models (baselines) are shown in Figure 6.

Figure 6 indicates that while our time-aware models trained with uniform sampling strategy generally perform better in most environments and have significantly better performance at low sampling rate (inference $\Delta t \geq 30$ms), they have lower success rates at small inference $\Delta t$ ($\Delta t \leq 2.5$ms) on in some environments, such as mw-assembly and mw-lever-pull. Regardless of the $\Delta t$ sampling strategy, the time-aware models have superior performance compared to the non time-aware baselines. Therefore, our time-aware model can be efficiently and effectively trained with any reasonable sampling strategy and is not only limited to log-uniform or uniform sampling. Deriving an optimal $\Delta t$ sampling strategy can be an interesting line of future work to achieve the highest performance on both small and large $\Delta t$ values. In the meanwhile, our log-uniform sampling strategy works well in practice, given our experimental results.

## C   ADDITIONAL COMPARISONS WITH NON TIME-AWARE BASELINES.

In this section, we extend Figure 5 and include additional results of other environments. Specifically, the learning curves of our time-aware model and the baseline non-time-aware TDMPC2 models evaluated on different inference $\Delta t$s are shown in Figure 7.

## D   COMPARISONS WITH MULTI TIME SCALE WORLD MODEL (MTS3).

As mentioned in Section 2, Multi Time Scale World Model (MTS3) (Shaj et al., 2023) is a closely related work with similar high-level motivation with our work: to model the world dynamics at multiple temporal levels. Specifically, MTS3 proposes a probabilistic approach to jointly learn the world dynamics at two temporal abstractions: task level (slow dynamics/timescale) and state level (fast dynamics/timescale). These 2 timescales are separately learned by two state space models (SSMs): $SSM^{fast}$ and $SSM^{slow}$, where $SSM^{fast}$ learns the dynamics evolving at original small timestep $\Delta t$ of the dynamical systems and $SSM^{fast}$ learn the slow dynamics evolving at $H\Delta t$. Although this approach also explicitly considered different temporal abstraction levels in learning the world dynamics, there are several critical differences between MTS3 compared to our work:

1. **Models vs Training method:** Shaj et al. (2023) proposes a model architecture to learn a world model with several discrete temporal abstraction levels. On the other hand, we proposed a *simple yet effective and efficient time-aware training method* that can be employed to train any world model architecture.

2. **Discrete vs continous timescales:** The original MTS3 currently only handle only 2 timescales: $\Delta t$ and $H\Delta t$, where both $\Delta t$ and $H$ is fixed in the training process. Although the MTS3 can be adapted to learn multiple timescales, the number of timescales is limited to a discrete value. Furthermore, the $SSM^{slow}$ (slow dynamic model) does not directly model state transition under large temporal gap $\Delta t$ (or low observation rate) but rather learns the task latents to guide $SSM^{fast}$ to long-horizon predictions. On the other hand, our time-aware approach can directly predict the future states $s_{t+\Delta t}$ under large $\Delta t$.

3. **Multi-step vs one-step prediction:** MTS3 considers the future prediction under large $\Delta t$ as a long-horizon prediction problem. Specifically, to predict $s_{t+\Delta t}$, MTS3 discretizes the long temporal gap into several smaller timesteps: $\Delta t = M\Delta t_{fast}$, where $\Delta t_{fast}$ is the original timestep size $SSM^{fast}$ is trained with and $M \in \mathbb{N}^+$. MTS3 then iteratively applies the model $M$ times to predict $s_{t+\Delta t}$. This timestep discretization approach has 2 critical limitations: **(1)** MTS3 cannot model state transitions under $\Delta t$ that is not divisible by $\Delta t_{fast}$ and **(2)** multi-step predictions are vulnerable to compounding errors, a well-known problem in long-horizon modeling. On the other hand, our model can directly

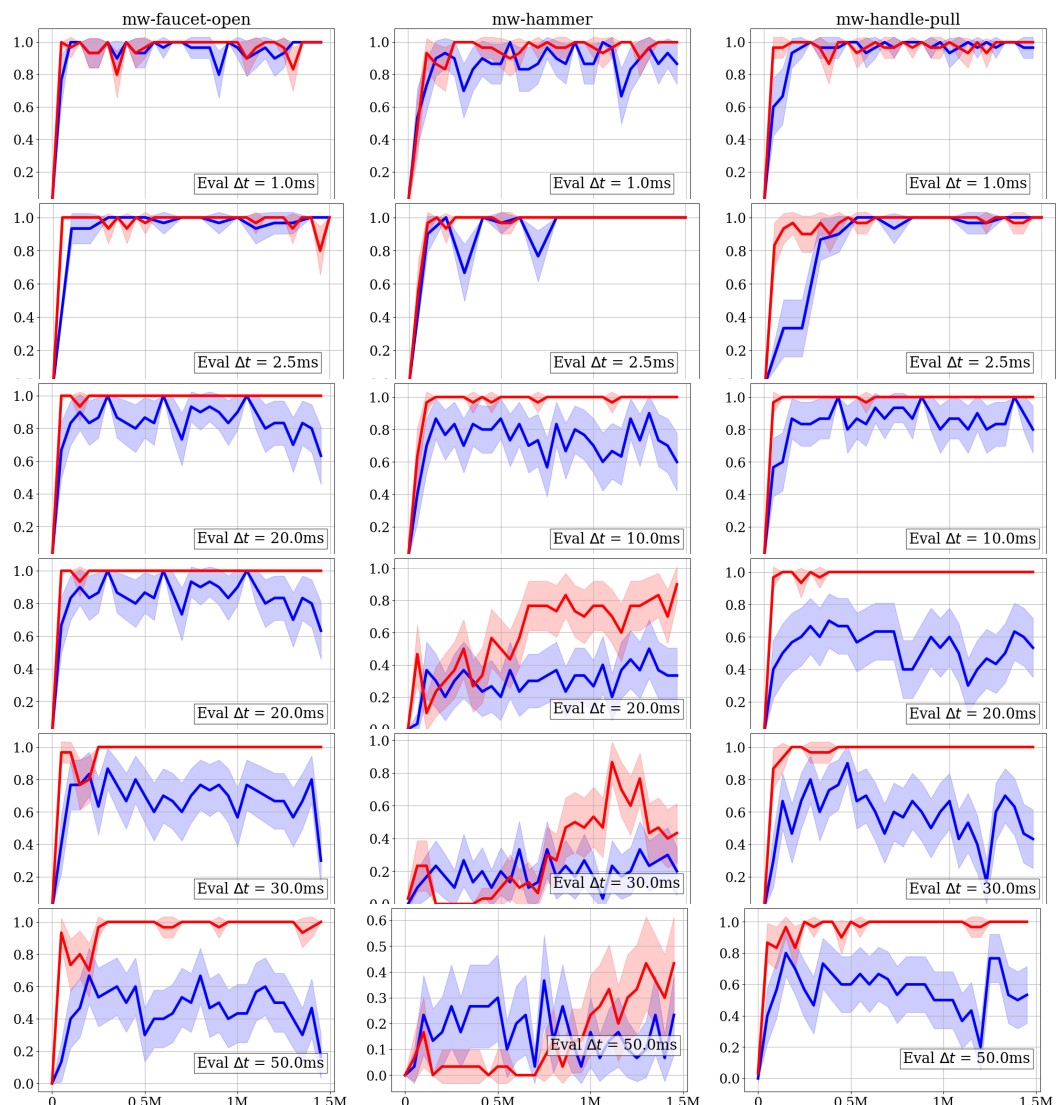

Figure 7: **Success Rate Curve under different evaluation time step sizes.** At each training step, the models are evaluated on various inference $\Delta t$s. Despite having to learn the dynamics under varying time step sizes, our time-aware model (in RED) still converges faster when evaluated on $\Delta t_{default} = 2.5$ms all tasks compared to the baseline trained only on $\Delta t_{default} = 2.5$ms (in BLUE). On large $\Delta t$'s, the time-aware model significantly outperforms the baseline with the same number of training steps while the baselines fail to converge. The mean and 95% confidence intervals success rate are plotted over 3 seeds, each with 10 evaluation episodes.

        predict the next state with a *one-step prediction*, effectively alleviating the compounding error problem.

4. **Inference efficiency:** Another disadvantage of multi-step prediction is inference inefficiency. In contrast, our time-aware model can efficiently predict long-term future states without sacrificing computational efficiency by using one-step prediction.

5. **Prediction-only vs Control:** As acknowledged by Shaj et al. (2023), the original MTS3 is strictly a prediction model. On the contrary, our time-aware model can be used efficiently with a planner to solve control problems.

In this section, we conduct empirical comparisons between MTS3 and our proposed time-aware model on the control problem, extending beyond the prediction-only scope in MTS3. First, MTS3

is trained with offline data consisting of $4 \times 10^6$ (4M transitions) collected from random trajectories (10%), half-trained policyś trajectories (20%), and trained expert policy trajectories (80%). Since MTS3 is strictly prediction-focused and is not designed for controls, we carefully combined MTS3 with MPPI planners and our world model's trained value and reward function. Implementation-wise, we replaced our dynamic model with MTS3 and kept all other components unchanged, including the planner (MPPI) and learned value and reward functions. This design ensures a fair comparison between the models, as any performance gap is attributed solely to the difference between MTS3 and our dynamic model. We kept the default hyper-parameter settings as in the original MTS3 paper and codebase. The results are shown in Figure 8. We will release the code source for the implementation of MTS3-MPPI in this experiment on GitHub when the paper is accepted.

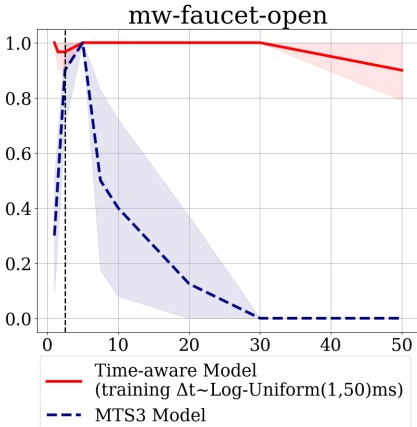

Figure 8: **Comparison with MTS3**. Average Episode Success Rate as a function of evaluation time steps (unit: millisecond). The dashed lines represent the default time step sizes ($\Delta t = 2.5$ms). The mean and $95\%$ confidence intervals are plotted over 3 seeds, each with 10 evaluation episodes. The baseline MTS3 models in DARK BLUE are trained with offline data with 4M transitions. Our time-aware models, highlighted in RED, are trained with 1.5M training steps.

The MTS3 inference stepping are adjusted such that when evaluated on $\Delta t_{eval} > \Delta t_{train}$, the model is applied $\Delta t_{eval}/\Delta t_{train}$ times ($\Delta t_{train}$ is the fast time step between $SSM^{fast}$'s observations). Figure 8 shows a rapid performance degradation of MTS3 as the evaluation timestep size increases, suggesting MTS3 also suffers from compounding error due to long-horizon prediction.

