# OpenReview forum: "Time-aware World Model:  Adaptive Learning of Task Dynamics"
_ICLR.cc/2025/Conference — Submitted to ICLR 2025_

### Official Review · Reviewer_hD4m · 2024-10-27

**Soundness:** 3
**Presentation:** 3
**Contribution:** 2
**Rating:** 6
**Confidence:** 3

**Summary:**

The paper introduces the Time-Aware World Model (TAWM), which adapts to the temporal dynamics of environments by conditioning on the time step size ($\Delta t$). Different from conventional models that use a fixed time step, TAWM trains across a diverse range of ∆t values, allowing it to capture both high- and low-frequency task dynamics. This approach addresses shortcomings in existing models, such as temporal resolution overfitting and inaccurate system dynamics when applied to real-world scenarios.

**Strengths:**

The time-aware mechanism significantly improves the emprical results compared to the non time-aware methods. The paper writing is improved and easy to understand.

**Weaknesses:**

1. The paper writing clearance and correctness should be improved. In Line 140, what does $\eta(\pi)$ mean? policy or expected return? And why $s_i$ could be sampled from a policy $\pi$?
2. What is the advantage of the proposed method compared to directly setting the simulation frequency 2x bigger than the real-world frequency? It seems that your proposed method and baselines all work under small $\Delta t$.
3. As depicted in Fig.5, your proposed method is not significantly superior to the baselines in empirical performance.

**Questions:**

Could you please provide more results of the comparisons between your proposed method and MTS3?

---

> ### Author Response · Authors · 2024-11-25
> **Response to reviewer hD4m**
>
> > The paper writing clearance and correctness should be improved. In Line 140, what does $\eta(\pi)$ mean? policy or expected return? And why $s_i$ could be sampled from a policy $\pi$?
>
> We apologize for the confusing description, especially in Line 140. It was a typo in Line 139 – specifically, we’d like to make a correction by changing “*obtain a policy or planner $\eta(\pi)$…*” to “*obtain a policy or planner $\pi$…*”
> In this context, $\eta(\pi)$ means the expected return for the policy $\pi$, which can be estimated by sampling trajectories using $\pi$.
>
> > What is the advantage of the proposed method compared to directly setting the simulation frequency 2x bigger than the real-world frequency? It seems that your proposed method and baselines all work under small $\Delta t$.
>
> Thank you for your question on the performance on real-world frequencies.  In our paper, we have carefully considered the real-world frequencies in each problem. Specifically, the largest Δt that we consider in our problem is Δt = 0.05s = 50ms, which means the frequency can be as low as 1/0.05 = 20Hz, which is lower than the typical robotics control frequency of 50-100Hz (or Δt ranging 10 to 20ms).
>
> As shown in **Appendix B, Fig. 6**, when we prioritize achieving high performance at low sampling frequencies, we can use uniform sampling strategy to sample training Δt’s, which results in near or 100% success rate at inference Δt = 50ms (or 20Hz), which is 2.5x to 5x larger than real-world Δt of 10 to 20ms. Regardless of the chosen sampling strategy, our time-aware model consistently outperforms the baseline models across different environments at different inference Δt’s.
>
> > As depicted in Fig.5, your proposed method is not significantly superior to the baselines in empirical performance.
>
> Please see our **General response A1**. We have added additional results to the **updated Figure 5** and **Figure 7 (Appendix C)** to better highlight the superior performance of our time-aware models over the non time-aware baselines.
>
> > This paper has no impressive strengths.
>
> Please see our **General response A2** where we highlighted the key contributions and impacts of our work. We hope that it can address your concerns about the novelty, contributions, and impacts of our work.

---

> > ### Comment · Reviewer_hD4m · 2024-12-02
> >
> > I appreciate the authors' detailed responses. The supplementary experiments improved the supports for the proposed claims. Especially, the experiment and additional explainations about related work MTS3 further strengthen this paper. I have increased my score to 5 and I am willing to further increase my score if you can provide more results versus MTS3. I hope you can submit the results before the end of the review period.

---

> ### Author Response · Authors · 2024-11-30
> **Responses to your questions and misunderstanding, plus revision with additional results requested**
>
> Dear reviewer hD4m,
>
> Thank you again for your comments to improve our paper's clarity and soundness. We have addressed your concerns as much as possible as shown in our previous response above, the **General Response**, and in the **revised paper**. We would be really grateful if you could take a look at our rebuttal and kindly let us know if you think that your concerns and questions are sufficiently addressed.  If not, please let us know what additional comments or questions you may have.  Kindly note, as summarize above:
>
> - We highlight our experiments on **NINE** very different tasks of MetaWorld benchmarks (Fig. 2). Our time-aware world model *consistently outperforms* baseline models by **SIGNIFICANT margins**.  (Ours achieves nearly 100% success rates across **all** time steps, in sharp contrast to baseline success rates, which drop quickly from 100% down to nearly 0%, as the time step increases from 1.0 msec up to 50 msec (please see Fig. 3, 4, 5).
>
> - This is the first work that shows a novel adaptive sampling strategy (by conditioning on varying time steps in observing samples for learning) that achieves a consistently high success task learning rates (**high 90% upto 100%**) for World Models. Our algorithm in adaptive sampling on time steps offers a far more robust strategy in learning and achieves a far more consistenly high task learning rates against most recent work like MTS3 (which only conditioned on 2 time scales: fast and slow). We also ran more comparison tasks (Fig. 2) and different action & observation rates and time steps, as shown in Fig. 3, 4, and 5.
>
> - This adaptive sampling framework conditioned on time step for observation rates is not limited or targeting solely on learning tasks in Robotics, but **ALL learning tasks involving dynamical systems**, such as traffic prediction, simulation for rapid prototyping and design, virtual try-on, and more. We will be happy to release our code to further research upon the accepted publication of this research.
>
> Please let us know if you have any additional concerns about our work.
>
> Best regards,
>
> The Authors.

---

> ### Author Response · Authors · 2024-12-02
> **Response to reviewer hD4m: Thank you for your response and evaluation!**
>
> Dear reviewer hD4m,
>
> We sincerely appreciate your time reviewing our rebuttal and the revised paper! Additionally, we would also like to express our sincere gratitude for the increase in your evaluation score. We are very happy to learn that our rebuttal and the additional results have addressed most of your concerns.
>
> To provide additional experiments on MTS3, we have conducted additional experiments with different long-term timescale settings for MTS3 by varying the $H$ values. We would like to provide additional comparisons between MTS3 ($H=3$), MTS3 ($H=11$), MTS3 ($H=33$), MTS3 ($H=50$), and our proposed method in the table below. The tested environment is `mw-faucet-open`, which is similar to current Figure 8 in **Appendix D** but with additional comparisons with MTS3 under varying $H$s.
>
> | Eval $\Delta t$ \| | MTS3 ($H=3$) \| | MTS3 ($H=11$) \| | MTS3 ($H=33$) \| | MTS3 ($H=50$) \|  | Our Method  |
> |-------------|:--------------:|:--------------:|:--------------:|:--------------:|:--------------:|
> | $1$ msec         | 60%  | 30%   | 100% | 100%  | 100%  |
> | $2.5$ msec      | 50%  | 90%   | 100% | 100%  | 97%    |
> | $5$ msec         | 70%  | 100% | 90%   | 100%  | 100%  |
> | $7.5$ msec      | 10%  | 50%   | 90%   | 90%    | 100%  |
> | $10$ msec       | 0%    | 40%   | 70%   | 100%  | 100%  |
> | $20$ msec       | 0%    | 10%   | 0%     | 70%    | 100%  |
> | $30$ msec       | 10%  | 0%     | 0%     | 30%    | 100%  |
> | $50$ msec       | 0%    | 0%     | 0%     | 0%      | 90%    |
>
> We observe that although increasing $H$ to model slower time dynamics tends to improve the performance of MTS3, the Time-Aware Model still performs significantly better than all MTS3 models, which is limited to learning only two timescales: $\Delta t$ and $H\Delta t$ as discussed in our revised paper's **Appendix D**.
>
> We would like to note that $H=11$ is the closest to the MTS3 authors' suggestion of setting $H=\sqrt{T}$, where $T$ is the episode length. In our experiments, $T=99$, and we chose $H=11 \approx \sqrt{99}$ to divide the episode into local SSM windows with equal lengths.
>
> Additionally, we also extended our experiments to `mw-basketball`, which has different underlying task dynamics and motion characteristics from those of `mw-faucet-open`. Our results for `mw-basketball` are shown in the table below:
>
> | Eval $\Delta t$ \| | MTS3 ($H=3$) \| | MTS3 ($H=11$) \| | MTS3 ($H=33$) \| | MTS3 ($H=50$) \| | Our Method  |
> |-------------|:--------------:|:--------------:|:--------------:|:--------------:|:--------------:|
> | $1$ msec         | 0%  | 0%   | 10% | 0%    | 76%    |
> | $2.5$ msec      | 0%  | 0%   | 0%   | 0%    | 97%    |
> | $5$ msec         | 0%  | 0%   | 0%   | 0%    | 100%  |
> | $7.5$ msec      | 0%  | 0%   | 0%   | 0%    | 100%  |
> | $10$ msec       | 0%  | 0%   | 0%   | 0%    | 84%    |
> | $20$ msec       | 0%  | 0%   | 0%   | 0%    | 100%  |
> | $30$ msec       | 0%  | 0%   | 0%   | 0%    | 97%    |
> | $50$ msec       | 0%  | 0%   | 0%   | 0%    | 27%    |
>
> In addition to the experiments on task success rate (%), we also investigated the runtime efficiency of MTS3 and compared it with the runtime efficiency of our model. We find that while our model's inference time is constant to the evaluation $\Delta t$ (0.04 to 0.06 seconds per step), MTS3's inference time scales linearly with $\Delta t$, on average requiring 2.45, 2.46, 4.86, 6.24, 10.56, 18.76, 56.84, and 98.75 seconds per step for $\Delta t$ = 1, 2.5, 5, 7.5, 10, 20, 30, 50, respectively. These results show that the Time-Aware Model does not only have higher success rates than MTS3 but is also more inference-efficient, which is important for control problems. We will make sure to update additional experimental results in new figures in the final paper revision.
>
> We hope that the additional experiments above help provide additional insights into the comparative performance between MTS3 and our model. If you have any additional questions or concerns, please kindly let us know!
>
> We again sincerely thank you for your time, support, and suggestions for our research, which are invaluable to improving our paper's clarity and strengths!
>
> Sincerely,
>
> The Authors

---

> > ### Author Response · Authors · 2024-12-03
> >
> > Dear reviewer hD4m,
> >
> > We appreciate your time and your patience. We would like to update you that we have conducted additional experiments on the performance of MTS3 across different long timescale settings by varying $H$ values. Furthermore, we included additional experiments on the `mw-basketball` task. Finally, we compared the inference runtime between MTS3 and our models on different evaluation $\Delta t$s. The results and analysis are carefully included in our previous response.
> >
> > Since the rebuttal period will be over in only a few hours, would you mind kindly taking a look at our new results and let us know if you have any further questions?
> >
> > We hope that the new results help make our paper stronger and can have an influence on your final evaluation.
> >
> > Sincerely,
> >
> > The Authors

---

### Official Review · Reviewer_NmDP · 2024-11-03

**Soundness:** 4
**Presentation:** 3
**Contribution:** 3
**Rating:** 8
**Confidence:** 4

**Summary:**

The paper proposes time step aware world models for handling real life distribution shifts with lower observation frequencies. It includes the time step as an additional input to the world model and uses log-uniform time step sampling during training to learn the world model for various observation frequencies. It shows impressive results on various control tasks within Meta-World, without any increase in the sample complexity.

**Strengths:**

The proposed strategy shows impressive performance when combined with TD-MPC2 on a variety of control tasks while using the same number of samples.
The proposed method can be combined with any existing MBRL algorithms such as TD-MPC2.

**Weaknesses:**

One of the motivations for the TD-MPC2 work was to create generalist agents, which can be trained in a multi-task setup and then can be easily fine tuned on any new task. However, no comparison with TD-MPC2 has been made for a multi-task setup.
A closely related work (as mentioned by the authors) on multi time scale world models (Shaj et al.) is missing as a comparison baseline in the experimental section.

**Questions:**

A precise description of how 4th order runge kutta method is used for integration is missing from section 4.1.2 or Algorithm 1. Can the authors please state it here as well as in the main paper for clarity?

---

> ### Author Response · Authors · 2024-11-25
> **Response to reviewer NmDP**
>
> > One of the motivations for the TD-MPC2 work was to create generalist agents, which can be trained in a multi-task setup and then can be easily fine tuned on any new task. However, no comparison with TD-MPC2 has been made for a multi-task setup.
>
> Thank you for your comments and suggestions! We agree that developing a multi-task generalist agent is the primary motivation behind TD-MPC2, and we anticipate that our method would also be effective for training such agents. However, our contribution goes beyond simply enhancing TD-MPC2 by introducing a time-aware element. In this paper, we highlight the significance of the temporal axis in world models, an aspect that has NOT been previously addressed.
>
> We believe that our proposed strategy is not limited to TD-MPC2 but can also be applied to other world dynamics models. We chose TD-MPC2 for our experiments because it is one of the latest world models, not because our approach is exclusive to it. Therefore, we argue that it is not strictly necessary to validate our method within multi-task settings to demonstrate its effectiveness. Please see our **General response A2** where we highlighted the key contributions and impacts of our work.
>
> In the revised version of the paper, we will clarify this point further and, where possible, aim to extend our evaluation to multi-task settings and other world models to strengthen our arguments.
>
> > A closely related work (as mentioned by the authors) on multi time scale world models (Shaj et al.) is missing as a comparison baseline in the experimental section.
>
> Although both our Time-Aware model and the Multi-Time Scale State Space World Model (MTS3) by Shaj et al. share a similar motivation of modeling the dynamics model under different time scales Δt’s, our core motivation and methodology differs substantially from MTS3, which is the reason why we only acknowledged the MTS3 paper as a related work but did not include it in the experimental section.  We would like to list the key differences of our proposed Time-Aware World Model with MTS3 below:
> 1. Although both MTS3 and our work share similar motivation, the core contribution of our work and MTS3 are different: while MTS3 proposes a novel architecture and modeling approach, we introduce a novel, simple, and efficient method to train any world model that can capture the underlying dynamics at different time step size Δt’s. Specifically, we find that by conditioning the dynamic models on Δt and simply varying Δt’s during the training process, the world model can effectively capture the unknown dynamics under different Δt’s **without requiring additional training steps or data** (please see **Figure 5, Figure 7 (Appendix C), and General response A1**). Since our main contribution is a training method, the time-aware training framework can be employed to train *any* world model architecture, including MTS3.
> 2. Although MTS3 also consider the on multi-time scale prediction problem, MTS3 primarily focuses on 2 time scales: short time-scale SSM (Δt) and long time-scale SSM (HΔt), where Δt and H is fixed for each model, with the goal of improving the accuracy of long-horizon prediction. As a result, their models are specialized in capturing the dynamics in 2 time scales: Δt and HΔt.
> On the other hand, our work can handle a wide range of Δt with a single-step prediction. Our motivation is not just long-horizon prediction, which involves multiple prediction steps, but single-step prediction for *varying* time step size Δt’s. The benefit of single-step prediction for varying Δt is the compatibility with real-world constraints, such as observation rates (e.g.: 60 fps) and control frequency (e.g.: 50Hz).
> 3. Methodology-wise, the MTS3 model is not explicitly conditioned on the Δt, which limits their performance to two timescales: Δt and HΔt for each trained model. On the other hand, by conditioning the dynamic model on the step size Δt, our model can quickly adapt to a wide range of Δt with single-step prediction.
>
> > A precise description of how 4th order runge kutta method is used for integration is missing from section 4.1.2 or Algorithm 1. Can the authors please state it here as well as in the main paper for clarity?
>
> We have added our precise description of the 4th-order Runge Kutta method for integration in **Appendix A**.

---

> ### Comment · Reviewer_NmDP · 2024-11-26
>
> Thanks for the response. While I do agree with the point about TD-MPC2, I am not convinced that MTS3 should not be included as a baseline. MTS3 as mentioned in the original paper could be easily extended to multiple time horizons. Moreover, as shown in the work, even a model trained for only two time steps ($\Delta t$ and $H \Delta t$) performs much better on other time scales, even beyond $H \Delta t$.
>
> The main comparison is indeed in between explicitly providing $t$ as input to the world model or using Bayesian inference for training a model which implicitly performs well on different time scales. I do not think this can be justified by saying MTS3 can be trained with $\Delta t$ as the input to the world model. Without the relevant baseline, I intend to keep my score.

---

> ### Author Response · Authors · 2024-11-27
> **Response to reviewer NmDP: comparison with MTS3**
>
> Thank you for your comments and suggestions! We will include additional comparisons with MTS3 as suggested and update the results in our final revised paper.

---

> > ### Comment · Reviewer_NmDP · 2024-11-28
> >
> > Thanks for the new experiments on MTS3. Can you please provide the MTS3 curves for different value of H - maybe 5, 7, 10 and 20? I think the current one is 3, which seems much shorter compared to the (1,50) ms interval?

---

> > > ### Author Response · Authors · 2024-11-29
> > > **Response to reviewer NmDP: addressing concerns about MTS3's H values**
> > >
> > > Dear reviewer NmDP,
> > >
> > > Thank you for taking the time to review our revised paper and for raising your concerns about the impacts of different $H$ values on MTS3's performance! We would like to clarify that in our experiment, we use $H=11$ (not $H=3$) as the hyperparameter for the MTS3 model. The reason we used $H=11$ is because the environments have an episode length of $T=99$. The authors of MTS3 suggested using $H=\sqrt{T}$, which is $\sqrt{99}\approx10$, and we chose $H=11$ to divide the episodes into equal-length local SSM windows.
> > >
> > > In addition to $H=11$, we also trained and experimented with MTS3 with $H=33$ (which also divides $99$). To ensure a fair comparison between our method and MTS3, we replaced our trained dynamic model component in our model with MTS3 and kept all other components unchanged, including the MPPI planner and the learned reward functions. Therefore, any performance gap between the two models is attributed solely to the difference between MTS3 and our dynamic model. We would like to refer to **Appendix D** for more detailed descriptions of the experiments. We summarize the performance between MTS3 ($H=11$), MTS3 ($H=33$), and our approach in the below table (measured in success rate):
> > >
> > > | Eval $\Delta t$ \| | MTS3 ($H=11$) \| | MTS3 ($H=33$) \|    | Our Method  |
> > > |-------------|:--------------:|:--------------:|:--------------:|
> > > | $1$ msec         | 30%  | 100% | 100% |
> > > | $2.5$ msec      | 90%  | 100% | 97% |
> > > | $5$ msec         | 100%  | 90% | 100% |
> > > | $7.5$ msec      | 50%  | 90% | 100% |
> > > | $10$ msec       | 40%  | 70%| 100%  |
> > > | $20$ msec       | 10%  | 0% | 100% |
> > > | $30$ msec       | 0%  | 0% | 100% |
> > > | $50$ msec       | 0%  | 0% | 90% |
> > >
> > > The results demonstrate that our method outperforms MTS3 in both settings. We hope that our clarification about the $H$ value and additional experiments of MTS3 ($H=33$) addresses your concern about the experiment. Additionally, besides the success rate (%), we also conducted further analysis of the inference runtime efficiency. We find that while our model's inference time is constant with respect to inference $\Delta t$ (0.04 to 0.06 seconds per step), MTS3's inference time scales linearly with $\Delta t$, requiring 2.45, 2.46, 4.86, 6.24, 10.56, 18.76, 56.84, and 98.75 seconds per step for $\Delta t$ = 1, 2.5, 5, 7.5, 10, 20, 30, 50, respectively.
> > >
> > > We sincerely appreciate your encouragement of our research potential. Your suggestions and questions are particularly meaningful for us to improve the clarity and strength of our paper. If you think that your concerns have been sufficiently addressed, we would be deeply grateful if you could consider raising the review rating.
> > >
> > > Best regards,
> > >
> > > The Authors

---

> ### Comment · Reviewer_NmDP · 2024-12-01
>
> Thanks for the additional results. I am happy to see that it outperforms MTS3 on control tasks. Based on this, I am happy to revise my rating.

---

> ### Author Response · Authors · 2024-12-01
> **Thank you for your reviews and support!**
>
> Dear Reviewer NmDP,
>
> Thank you so much again for your helpful comments and suggestions to make this paper a much stronger publication!
> We appreciate your insightful comments, and we will make sure to incorporate the results in our discussions into the final revision of our paper!
>
> Best regards,
>
> The Authors

---

### Official Review · Reviewer_8dTB · 2024-11-04

**Soundness:** 3
**Presentation:** 4
**Contribution:** 1
**Rating:** 5
**Confidence:** 4

**Summary:**

This paper emphasizes the importance of incorporating temporal information $\Delta t$ into dynamic modeling and introduces a mixture-of-time-step training framework that learns task dynamics across multiple frequencies. The authors' motivation stems from the existence of multi-scale dynamical systems, where each subsystem may operate at a unique frequency, and from the Nyquist-Shannon sampling theorem, which implies that lower sampling frequencies reduce performance, while higher sampling frequencies improve accuracy but increase sample complexity and reduce learning efficiency. Their approach involves conditioning all components of a world model, except the encoder, on $\Delta t$ and using the Euler or RK4 integration method to reformulate the dynamic model. The authors experimentally validate their method on several control tasks from the Meta-World suite, using TD-MPC2 as the baseline. They demonstrate that their time-aware modifications to TD-MPC2 yield superior performance over the baseline when evaluated across varying frequencies at test time, all while maintaining the same sample efficiency.

**Strengths:**

- The paper is well-written and clear, with strong motivation and thorough explanations of each aspect of their approach.

- The proposed idea is innovative and promising, with potential for significant impact on real-world applications.

**Weaknesses:**

- Since the core framework relies heavily on pre-existing world models and the main contribution is incremental, I expected to see larger-scale experiments in more complex environments, such as real robots or video games, to better illustrate the approach's applicability. In its current form, the paper lacks sufficient novelty to fully engage readers. I recommend that the authors either incorporate experiments in more complex environments to demonstrate real-world applicability or create more challenging scenarios, such as those requiring adaptation to varying $\Delta t$ within the same episode.


- The authors claim that smaller frequencies increase sample complexity and are therefore lead to inefficient learning. However, the paper lacks theoretical or experimental evidence to explain why this is true. To strengthen this claim and further support the motivation for time-aware world models, I suggest adding smaller frequencies to the experiment in Figure 4 and comparing their sample complexity with that of the time-aware model. This addition would provide valuable insight into the efficiency benefits of the proposed approach.

**Questions:**

- Is there a specific reason why lower frequencies are missing in Figure 4? For instance, I was interested in seeing the baseline's performance with $\Delta t =0.1$ ms, as this frequency is used during the training of the time-aware model.

- Section 3.2.3 appears to be a mitigation for the issue described in 3.2.2. Would it be reasonable to combine these sections, or are they based on distinct motivations?

- Since TD-MPC2 assumes an underlying MDP structure, I suggest replacing $o_t$ with $s_t$ in Section 4.1 to avoid confusion with observations in a POMDP setting.

---

> ### Author Response · Authors · 2024-11-25
> **Response to reviewer 8dTB**
>
> > Since the core framework relies heavily on pre-existing world models and the main contribution is incremental, I expected to see larger-scale experiments in more complex environments, such as real robots or video games, to better illustrate the approach's applicability. In its current form, the paper lacks sufficient novelty to fully engage readers. I recommend that the authors either incorporate experiments in more complex environments to demonstrate real-world applicability or create more challenging scenarios, such as those requiring adaptation to varying $\Delta t$ within the same episode.
>
> Thank you for your comments and suggestions! We would like to emphasize that **our core contribution is the time-aware world model training framework and not architecture contribution**. Since our main contribution is a time-aware training framework, our method is **model-agnostic**, which **can be easily employed to train any world model architecture**. Please see our **General response A2** where we explain the novelty and contributions of our work.
>
> More challenging environments, like the suggested one where Δt varies within the same episode, would be even more powerful to show the effectiveness of our approach generalized to a dynamical system with *changing frequencies* (thereby requiring varying Δt).  Such scenarios, however, are not common occurrences and have not been shown in recent publications or encountered in typical real-world applications.  We will add such challenging environments in the revised version of the paper. However, as a theoretical paper that deals with the (previously ignored) temporal elements of the world model, we’d like to note that we have already done extensive verification and comparison using several diverse sets of benchmarks with inherently different fundamental frequencies that are not known in advance. Our time-aware world model can automatically achieve the best possible performance without any assumption on the dynamical systems.
>
> > The authors claim that smaller frequencies increase sample complexity and are therefore lead to inefficient learning. However, the paper lacks theoretical or experimental evidence to explain why this is true. To strengthen this claim and further support the motivation for time-aware world models, I suggest adding smaller frequencies to the experiment in Figure 4 and comparing their sample complexity with that of the time-aware model. This addition would provide valuable insight into the efficiency benefits of the proposed approach.
>
> Thank you for the suggestion! As mentioned in our response to Reviewer fXm1, we observed that using the highest possible sampling frequency often resulted in suboptimal training efficiency for the world dynamics model. This is reflected in Figure 3, where our time-aware model outperformed the baseline after only 1.5M training steps, while the baseline model took 2M steps to achieve inferior results. Additionally, please see our **General response A1**.
>
> We believe this is due to the increased sample complexity caused by higher sampling frequencies, which lead to generating unnecessarily redundant state transition data. These redundant samples provide limited additional information about the system dynamics, making training less efficient.
>
> We will highlight this point more clearly in the revised version of the paper and include additional experimental results to further demonstrate the efficiency of our approach.
>
> > Is there a specific reason why lower frequencies are missing in Figure 4? For instance, I was interested in seeing the baseline's performance with $\Delta t=0.1$ms, as this frequency is used during the training of the time-aware model.
>
> Thank you for pointing out such detail.  We’d like to make a typo correction that we trained the time-aware model with Δt ranging from $[1,50]$ms instead of $[0.1,50]$ms. We have made the correction in the revised paper.
>
> > Section 3.2.3 appears to be a mitigation for the issue described in 3.2.2. Would it be reasonable to combine these sections, or are they based on distinct motivations?
>
> In Section 3.2, we first introduce the nature of dynamical systems (3.2.1) and then present a theorem that describes the sampling conditions required to reconstruct the signals of such systems (3.2.2). Finally, we propose our sampling strategy for reconstructing signals based on these conditions (3.2.3). While these subsections share a common underlying motivation, we will consider combining them and streamlining the discussion to improve clarity in the revised version of the paper.
>
> > Since TD-MPC2 assumes an underlying MDP structure, I suggest replacing $o_t$ with $s_t$ in Section 4.1 to avoid confusion with observations in a POMDP setting.
>
> Thank you for pointing it out, we will change the notation in the revised version of the paper to avoid confusion.

---

> ### Comment · Reviewer_8dTB · 2024-11-27
>
> Thank you for the clarification. However, I still believe the core idea of the method is straightforward. Without addressing more complex settings/applications or providing a solid theoretical foundation, the paper's contribution remains limited. Therefore, I will keep my current score.

---

### Official Review · Reviewer_fXm1 · 2024-11-04

**Soundness:** 2
**Presentation:** 3
**Contribution:** 2
**Rating:** 3
**Confidence:** 4

**Summary:**

For model-based RL in continuous time domains, the paper proposes to learn a time-aware world model whose components depend on the sampling frequency. In the training phase, the time-aware world model is trained with randomly varying time steps. In numerical experiments, the proposed method is evaluated under different time steps and show better performance compared with the baseline trained at the fixed time step.

**Strengths:**

- When approximating a continuous-time dynamical system with a sampled model, having a time-step dependent world model makes a lot of sense to better fit the actual dynamics. The paper proposes such a time-aware world model with the potential to improve model-based RL performance.

**Weaknesses:**

- The motivation referring to the Nyquist-Shannon sampling theorem does not seem to be consistent with the observation of numerical experiments. From the sampling theorem, as long as the sampling frequency is high enough, there won't be any information loss. But the numerical experiments suggest that the non time-aware model performs poorly even when the sampling frequency is the highest. This inconsistency makes the connection to the sampling theorem questionable.

- During the training phase, the sampling step is randomly selected from a log-uniform distribution. It is suggested that this distribution helps stabilize the training process, but no theoretical nor numerical analysis is provided. Some ablation studies using different sampling distribution might provide some insights to the choice.

- In the numerical evaluation, performance is compared across different observation rate. Given any time-step, the time-aware model can provide the appropriate prediction by conditioning on the time-step. But for the model trained with a fixed time-step, the information of the observation time-step does not seem to be adjusted. For example, for a model trained with $\Delta t=1$ms, when evaluating at $\Delta t=2$ms, instead of applying the model once, one might want to apply the model twice to have a more accurate prediction given the knowledge of the doubled time-step. Without doing some kind of adjustments for the baseline models make the fairness of comparisons questionable.

- When trained and evaluated with the same time-step, Figure 5 shows similar performance between the baseline model and the time-aware model. This makes the effectiveness of the proposed methods questionable when the goal is just to achieve good performance. Maybe under some scenarios where the training time-step and evaluation time-step are different, the proposed method might make more sense.

**Questions:**

- It would be great if those points listed in the weaknesses above could be addressed.

---

> ### Author Response · Authors · 2024-11-25
> **Response to reviewer fXm1 (1/2)**
>
> > The motivation referring to the Nyquist-Shannon sampling theorem does not seem to be consistent with the observation of numerical experiments. From the sampling theorem, as long as the sampling frequency is high enough, there won't be any information loss. But the numerical experiments suggest that the non time-aware model performs poorly even when the sampling frequency is the highest. This inconsistency makes the connection to the sampling theorem questionable.
>
> Thank you for your thorough comments! However, we respectfully disagree with the claim that the Nyquist-Shannon Sampling Theorem does not align with our observations. As you noted, the theorem establishes a necessary condition for reconstructing a signal without information loss. However, it does not imply that reconstructing the signal at the highest possible sampling frequency is the most efficient approach for all applications.
>
> In fact, in our experiments, we observed that using the highest possible sampling frequency often resulted in suboptimal training efficiency for the world dynamics model. To explain this intuitively, consider a dynamical system dominated by low-frequency signals (e.g., a scene where an object moves very slowly). Sampling state transitions at a very high frequency in such a system generates a large amount of redundant data, as most transitions are repetitive and do not contribute meaningful information about the system’s core dynamics. Training the model with these redundant transitions makes the process less efficient. Instead, reducing the sampling frequency ensures that the transitions capture more meaningful variations, allowing the dynamics model to be trained more effectively.
>
> That said, it is important to ensure the sampling frequency does not drop below the minimum required by the Nyquist-Shannon Theorem to avoid information loss. Since the minimum signal frequency is typically unknown in practice, we adopted a randomized sampling approach in our experiments to strike a balance between capturing meaningful dynamics and minimizing redundancy.
>
> In summary, we believe the suboptimal performance of the non-time-aware model using high sampling frequencies arises from this efficiency issue. We will provide a clearer explanation of this reasoning in the revised version of the paper.
>
> > During the training phase, the sampling step is randomly selected from a log-uniform distribution. It is suggested that this distribution helps stabilize the training process, but no theoretical nor numerical analysis is provided. Some ablation studies using different sampling distribution might provide some insights to the choice.
>
> Thank you for your comments about the Δt log-uniform sampling strategy during the training process.  First, **we’d like to clarify that the sampling strategy for training Δt can be any reasonable sampling strategy and is not limited to log-uniform**. The main motivation for using log-uniform sampling distribution is because log-uniform distribution allows us to collect samples at varying Δt’s (or varying frequency) during the training process, allowing the model to achieve better performance at smaller inference Δt’s (i.e. Δt’s close to Δtdefault = 2.5ms). In this context, “more stable learning process” means that the returns curve converges faster to the optimal performance (100% success rate) **when evaluated on $Δt_{default}$ = 2.5ms**. However, we want to emphasize that while we prefer log-sampling strategy because we aim to have a balanced model performance on not only large but also small Δt’s , **depending on the goals, the time-aware model should be flexible with any reasonable choice of Δt sampling strategy**, including but not limited to uniform sampling, which is better for achieving high-performance on low sampling rate (or large Δt) because large $Δt > Δt_{default}=2.5ms$ are sampled much more frequently (please see **Appendix B, Fig. 6**).
>
> To compare the log-uniform sampling strategy to more common strategies such as uniform sampling, we added an ablation study by training time-aware models using Δt ~ Uniform(1, 50) ms and compared them with the corresponding time-aware models with log-uniform sampling strategy. The results are shown in **Appendix B, Fig. 6**. While our time-aware models trained with uniform sampling strategy generally perform well in most environments and have significantly better performance at low sampling rate (inference Δt ≥ 30ms), they have lower success rates at small inference Δt (Δt ≤ 2.5ms) on mw-assembly. **Appendix B** shows that **our time-aware model can be efficiently and effectively trained with any reasonable sampling strategy and is not only limited to log-uniform or uniform sampling**. Deriving an optimal Δt sampling strategy can be an interesting line of future work to achieve the highest performance on both small and large Δt values.  In the meanwhile, our log-uniform sampling strategy works well in practice, given our experimental results.

---

> ### Author Response · Authors · 2024-11-25
> **Response to reviewer fXm1 (2/2)**
>
> > In the numerical evaluation, performance is compared across different observation rate. Given any time-step, the time-aware model can provide the appropriate prediction by conditioning on the time-step. But for the model trained with a fixed time-step, the information of the observation time-step does not seem to be adjusted. For example, for a model trained with $\Delta t=1$ ms, when evaluating at $\Delta t=2$ms, instead of applying the model once, one might want to apply the model twice to have a more accurate prediction given the knowledge of the doubled time-step. Without doing some kind of adjustments for the baseline models make the fairness of comparisons questionable.
>
> Thank you for raising your concern about the evaluation fairness.  We have **updated Figure 3** with additional comparisons to the baseline models with adjusted inference stepping according to the suggestion.
>
> To explain our original decision to use only single-step prediction for the non-time-aware model, the main purpose is to emphasize the limitation of non-time-aware models across different Δt’s by showing their failures to generalize to the state transition under $\Delta t_{eval} \neq \Delta t_{train}$ in a one-step prediction. For example, Figures 3 and 5 show that when the baseline models are trained using fixed $\Delta t_{train}=2.5$ms, the models perform well when the inference $\Delta t_{eval}=2.5$ms, but the performance degrades severely when $\Delta t_{eval}$ deviates from $\Delta t_{train}$. On the other hand, the time-aware models can accurately capture the state transitions under different Δt in a single-step prediction.
>
> We find that with inference adjustment, the performance of the baseline model not only did not improve but degraded. This is due to the well-known compounding error issue [1,2,3], as adjusting the inference step for higher inference Δt means expanding the horizon by a factor of $\Delta t_{eval} / \Delta t_{train}$, where $\Delta t_{train}$ is the time step size that the non-time-aware model is trained on. With increased horizon, the compounding error becomes more severe, degrading the effectiveness of the planner and thus the success rate significantly.
>
> On the other hand, using a mixture of $\Delta t$'s to train the time-aware model offers two advantages:
> 1. The time-aware model be easily used and adapted on different tasks under different observation rate $\Delta t$'s .
> 2. The time-aware model uses one-step prediction, overcoming the compounding error problem and avoiding introducing additional computational overhead.
>
> > When trained and evaluated with the same time-step, Figure 5 shows similar performance between the baseline model and the time-aware model. This makes the effectiveness of the proposed methods questionable when the goal is just to achieve good performance. Maybe under some scenarios where the training time-step and evaluation time-step are different, the proposed method might make more sense.
>
> Please see our global response A1.
>
> **References:**
> 1. Lambert, Nathan, Kristofer Pister, and Roberto Calandra. "Investigating compounding prediction errors in learned dynamics models." arXiv preprint arXiv:2203.09637 (2022).
> 2. Clavera, Ignasi, et al. "Model-based reinforcement learning via meta-policy optimization." Conference on Robot Learning. PMLR, 2018.
> 3. Wang, Tingwu, et al. "Benchmarking model-based reinforcement learning." arXiv preprint arXiv:1907.02057 (2019).

---

> ### Author Response · Authors · 2024-11-30
> **Responses to your questions and misunderstanding, plus revision with additional results as requested**
>
> Dear reviewer fXm1,
>
> Thank you again for your thorough comments, and we truly appreciate your invaluable insights and suggestions to improve our paper's clarity and soundness. We have addressed your concerns as much as possible as shown in our two previous responses, the **General Response**, and in the **revised paper**. We would be really grateful if you could take a look at our rebuttal and kindly let us know if you think that your concerns and questions are sufficiently addressed.  Kindly note, as summarize above:
>
> - We highlight our experiments on **NINE** very different tasks of MetaWorld benchmarks (Fig. 2). Our time-aware world model *consistently outperforms* baseline models by **significant margins**. (Ours achieves nearly 100% success rates across all time steps, in sharp contrast to baseline success rates, which drop quickly from 100% down to nearly 0%, as the time step increases from 1.0 msec up to 50 msec,(see Fig. 3, 4, 5).
>
> - This is the first work that shows a novel adaptive sampling strategy (by conditioning on varying time steps in observing samples for learning) that achieves a consistently high success task learning rates (**high 90% upto 100%**) for World Models. Our algorithm in adaptive sampling on time steps offers a far more robust strategy in learning and achieves a far more consistenly high task learning rates against most recent work like MTS3 (which only conditioned on 2 time scales: fast and slow). We also ran more comparison tasks (Fig. 2) and different action & observation rates and time steps, as shown in Fig. 3, 4, and 5.
>
> - This adaptive sampling framework conditioned on time step for observation rates is not limited or targeting solely on learning tasks in ROBOTICS, but **ALL learning tasks involving dynamical systems**, such as traffic prediction, simulation for rapid prototyping and design, virtual try-on, and more. We will be happy to release our code to further research upon the accepted publication of this research.
>
> Please let us know if you have any additional concerns about our work.
>
> Best regards,
>
> The Authors.

---

> > ### Author Response · Authors · 2024-12-02
> >
> > Dear reviewer fXm1,
> >
> > We thank you again for your detailed comments and suggestions. Since it is only 15 hours until the end of the rebuttal period, could you please confirm if you have reviewed our responses and the revised paper? Please kindly let us know whether our rebuttal and the revised paper with additional comparisons have influenced your evaluation.
> >
> > We sincerely appreciate your time in reviewing our work. It is truly meaningful for us to have your reviews to further improve our paper. If you feel that our rebuttal has adequately addressed your questions and concerns, we would be grateful if you could consider increasing the review rating.
> >
> > Sincerely,
> >
> > The Authors

---

> ### Author Response · Authors · 2024-12-02
> **Your comments and response, please?**
>
> Dear Reviewer fXM1,
>
> We'd sincerely appreciate your time to review our rebuttal and the revised paper! We would be grateful if you could consider increasing your evaluation if most of your concerns are addressed after reviewing our responses.
>
> To provide additional experiments on MTS3, we have conducted additional experiments with different long-term timescale settings for MTS3 by varying the $H$ values. We would like to provide additional comparisons between MTS3 ($H=3$), MTS3 ($H=11$), MTS3 ($H=33$), MTS3 ($H=50$), and our proposed method in the table below:
>
> | Eval $\Delta t$ \| | MTS3 ($H=3$) \| | MTS3 ($H=11$) \| | MTS3 ($H=33$) \| | MTS3 ($H=50$) \|  | Our Method  |
> |-------------|:--------------:|:--------------:|:--------------:|:--------------:|:--------------:|
> | $1$ msec         | 60%  | 30%   | 100% | 100%  | 100%  |
> | $2.5$ msec      | 50%  | 90%   | 100% | 100%  | 97%    |
> | $5$ msec         | 70%  | 100% | 90%   | 100%  | 100%  |
> | $7.5$ msec      | 10%  | 50%   | 90%   | 90%    | 100%  |
> | $10$ msec       | 0%    | 40%   | 70%   | 100%  | 100%  |
> | $20$ msec       | 0%    | 10%   | 0%     | 70%    | 100%  |
> | $30$ msec       | 10%  | 0%     | 0%     | 30%    | 100%  |
> | $50$ msec       | 0%    | 0%     | 0%     | 0%      | 90%    |
>
> We observe that although increasing $H$ to model slower time dynamics tends to improve the performance of MTS3, the Time-Aware Model still performs significantly better than all MTS3 models, which is limited to learning only two timescales: $\Delta t$ and $H\Delta t$ as discussed in our revised paper's **Appendix D**.
>
> We would like to note that $H=11$ is the closest to the MTS3 authors' suggestion of setting $H=\sqrt{T}$, where $T$ is the episode length. In our experiments, $T=99$, and we chose $H=11 \approx \sqrt{99}$ to divide the episode into local SSM windows with equal lengths.
>
> In addition to the experiments on task success rate (%), we also investigated the runtime efficiency of MTS3 and compared it with the runtime efficiency of our model. We find that while our model's inference time is constant to the evaluation $\Delta t$ (0.04 to 0.06 seconds per step), MTS3's inference time scales linearly with $\Delta t$, on average requiring 2.45, 2.46, 4.86, 6.24, 10.56, 18.76, 56.84, and 98.75 seconds per step for $\Delta t$ = 1, 2.5, 5, 7.5, 10, 20, 30, 50, respectively. These results show that the Time-Aware Model does not only have higher success rates than MTS3 but is also more inference-efficient, which is important for control problems. We will make sure to update additional experimental results in new figures in the final paper revision.
>
> We hope that the additional experiments above help provide additional insights into the comparative performance between MTS3 and our model. If you have any additional questions or concerns, please kindly let us know!
>
> We again sincerely thank you for your time and thorough reviews of our paper, which are very helpful for us to improve our paper's clarity and strengths!
>
> Sincerely,
>
> The Authors

---

### Official Review · Reviewer_mZgr · 2024-11-09

**Soundness:** 3
**Presentation:** 3
**Contribution:** 2
**Rating:** 6
**Confidence:** 3

**Summary:**

The authors propose an approach to address the issue of differing time step sizes $\Delta t$ between training and test phases, which can lead to a mismatch in dynamics. They propose a training scheme that randomly samples $\Delta t$ as inputs and update the model that explicitly conditions on $\Delta t$. They show that empirically this approach outperforms the baseline method, which was trained with a fixed time step size, when tested with a different $\Delta t$.

**Strengths:**

- This paper considers a novel problem setting which is also practically important, especially for robotics control tasks.

- The paper is well written. The motivation is well-stated and the methodology is presented clear.

- The proposed algorithm is evaluated on multiple tasks. And the empirical results are supportive of the main claim of the paper.

**Weaknesses:**

- I would suggest to add a comparison between the proposed approach with works about Robust RL (e.g. Pinto, Lerrel, et al. "Robust adversarial reinforcement learning." International conference on machine learning. PMLR, 2017). In my understanding, the mismatch in the time step size is one specific instance of the dynamics mismatch. I wonder if the proposed approach can get better performance by explicitly conditioning on $\Delta t$ only.

- I wonder if the log-uniform distribution is the best way to sample $\Delta t$. A justification for this would be helpful.

**Questions:**

Please check the above Weaknesses section.

---

> ### Author Response · Authors · 2024-11-25
> **Response to reviewer mZgr**
>
> > I would suggest to add a comparison between the proposed approach with works about Robust RL (e.g. Pinto, Lerrel, et al. "Robust adversarial reinforcement learning." International conference on machine learning. PMLR, 2017). In my understanding, the mismatch in the time step size is one specific instance of the dynamics mismatch. I wonder if the proposed approach can get better performance by explicitly conditioning on $\Delta t$ only.
>
> Thank you for bringing this work to our attention! We agree that our approach shares a similar spirit with robust RL methods, particularly the suggested RARL paper, in addressing the mismatch between training and testing environments. However, we would like to clarify that handling such mismatches is only one of the main inspirations behind developing the Time-Aware World Model for RL; it is not our sole objective.  In fact, as discussed in the Introduction, our motivation stems from the observation that dynamical systems are composed of signals with *varying* frequencies. Consequently, relying on previous world models that use a fixed time step without considering these temporal components is not an ideal approach. Addressing the mismatch problem is just one of the many benefits offered by our Time-Aware World Model. In the revised version of the paper, we will include this line of work in the related work section and provide more detailed comparisons with our approach.
>
> > I wonder if the log-uniform distribution is the best way to sample $\Delta t$. A justification for this would be helpful.
>
> Thank you for your comments about the Δt log-uniform sampling strategy during the training process.  First, **we’d like to clarify that the sampling strategy for training Δt can be any reasonable sampling strategy and is not limited to log-uniform**. The main motivation for using log-uniform sampling distribution is because log-uniform distribution allows us to collect samples at varying Δt’s (or varying frequency) during the training process, allowing the model to achieve better performance at smaller inference Δt’s (i.e. Δt’s close to Δtdefault = 2.5ms). In this context, “more stable learning process” means that the returns curve converges faster to the optimal performance (100% success rate) **when evaluated on $Δt_{default}$ = 2.5ms**. However, we want to emphasize that while we prefer log-sampling strategy because we aim to have a balanced model performance on not only large but also small Δt’s , **depending on the goals, the time-aware model should be flexible with any reasonable choice of Δt sampling strategy**, including but not limited to uniform sampling, which is better for achieving high-performance on low sampling rate (or large Δt) because large $Δt > Δt_{default}=2.5ms$ are sampled much more frequently (please see **Appendix B, Fig. 6**).
>
> To compare the effectiveness of the log-uniform sampling strategy to more common strategies such as uniform sampling, we added an ablation study by training time-aware models using Δt ~ Uniform(1, 50) ms and compared them with the corresponding time-aware models with log-uniform sampling strategy. The results are shown in **Appendix B, Fig. 6**. We can observe that while our time-aware models trained with uniform sampling strategy generally perform well on most environments and have significantly better performance at low sampling rate (inference Δt ≥ 30ms), they have lower success rate at small inference Δt (Δt ≤ 2.5ms) on mw-assembly. **Appendix B** shows that **our time-aware model can be efficiently and effectively trained with any reasonable sampling strategy and is not only limited to log-uniform or uniform sampling**. Deriving an optimal Δt sampling strategy can be an interesting line of future work to achieve the highest performance on both small and large Δt values.  In the meanwhile, our log-uniform sampling strategy works well in practice, given our experimental results.

---

> > ### Comment · Reviewer_mZgr · 2024-11-27
> >
> > Thanks for the response! While the authors have addressed most of my concerns, I am still concerned about whether existing approaches (such as Robust RL methods) can already perform well under the current experimental setup. Therefore, without including such methods as baselines, I prefer to maintain my score.

---

> > > ### Author Response · Authors · 2024-11-30
> > > **Response to Reviewer mZgr: regarding the baseline comparisons**
> > >
> > > Dear reviewer mZgr,
> > >
> > > We are happy to learn that most of your concerns have been addressed! Your suggestion to further include other relevant existing approaches as baselines beyond non-time-aware TDMPC2 is undoubtedly valuable to strengthen our paper. Could you please confirm if you have reviewed our response in **Summary of Rebuttal** and the revised paper? Please kindly let us know whether our response in **Summary of Rebuttal** and the revised paper with additional comparisons with MTS3 (in **Appendix D**) have sufficiently addressed your concerns about baseline comparisons.
> > >
> > > We sincerely appreciate your time reviewing our research and your comments to improve our paper, which is truly meaningful to improve the quality of our work.
> > >
> > > Best regards,
> > > The Authors

---

> > > > ### Comment · Reviewer_mZgr · 2024-11-30
> > > >
> > > > Thanks for the detailed comments and additional experiments. I will increase my score.

---

> > > > > ### Author Response · Authors · 2024-11-30
> > > > > **Response to reviewer mZgr: Thank you for your review and evaluation**
> > > > >
> > > > > Thank you very much for your invaluable support of our work! We sincerely appreciate your time and your feedback, and we are honored to address your concerns and questions. We will make sure to integrate your insights and suggestions in the final revision of our paper.

---

### Author Response · Authors · 2024-11-25
**General responses to common concerns from the reviewers**

We appreciate all the reviewers’ comments. Here we provide answers to the common questions from reviewers.

**Q1) Unclear advantage over the baseline in Figure 5 (Reviewer fXm1, hD4m)**

**A1)** Reviewers questioned the clear time-aware model’s advantage in Figure 5 of our time-aware model over the non time-aware baselines.  As shown in Figure 5, the reward curves of the time-aware models may only appear to be converging marginally faster, when compared to the baselines with the default timestep of Δt=2.5ms. However, we would like to emphasize that **these results actually show the advantage of our model over the baseline in terms of both performance and efficiency**: the time-aware model (trained on varying Δt’s) outperforms the baseline (trained only on Δt=2.5ms) when evaluated on different Δt’s (Fig. 3 and Fig. 4) while still converge noticeably faster than the baseline when evaluated on inference Δt=2.5ms (the exact Δt for which the baseline was specifically trained on, so the best performance of the baseline is expected at this exact timestep).  Yet we observed slightly superior performance of our Time-Aware model over the exact trained case for the baseline, with clearly superior performance on other varying Δt’s.  We have run extra experiments and included additional results **updated Figure 5** and **Appendix C** as additional evidence on the generalization of our claim.

Specifically, we included additional reward curves evaluated on different inference Δt’s. Implementation-wise, we saved intermediate model weights during the training process, and evaluated such intermediate model weights on different inference Δt’s to obtain the reward curves (i.e: different figures show the return curve of the same model trained in the same process but evaluated on different inference Δt’s). As shown in Fig. 5 and Appendix C, we can observe clear performance advantages of the time-aware model over the baselines when evaluated on various inference Δt’s, especially Δt>2.5ms while using the same number of training steps (1.5M). This behavior is observed consistently across different environments. These results show that the **time-aware model is able to outperform the non time-aware model across different inference Δt’s in the same number of training steps (or without requiring additional training data)**.

---

**Q2) No impressive strength and the main contribution is incremental. (Reviewer 8dTB, hD4m)**

**A2)**  First, we would like to emphasize that our **core contribution is the time-aware world model training framework, not the architecture contribution**. Particularly, we propose **a simple and highly efficient training method** by:
1. **Explicitly conditioning the dynamics model on time step size Δt**, one of the most important quantities in any dynamical system yet overlooked by the world model and reinforcement learning community.
2. **Train the world dynamic model on a mixture of Δt’s by randomly sampling Δt during the training process**.

To the best of our knowledge, **despite the simplicity of our method, our work is the first to propose such a time-aware training framework**, which is shown in ***Figure 3, 4, 5, 6, 7*** to consistently improve world model’s performance on varying inference time step sizes across different tasks without requiring additional training data, thus enhancing sample efficiency.

Since our main contribution is a time-aware training framework, our method is model-agnostic, which **can be easily employed to train any world model architecture**.

One important strength of our work is that the dynamic model is explicitly conditioned on Δt and is trained to directly predict the next state $s_{t+Δt}$ in **one-step prediction**. The main advantages of one-step prediction conditioned on Δt are:
1. Since our model can predict the next state under large Δt in a single step, it does not introduce additional computational overhead caused by multi-step predictions. As a result, it is compatible with real-world constraints of observation rate (e.g: 60fps) and/or control frequency (e.g: 120Hz).
2. Another benefit of single-step prediction is that we can avoid compounding error problems, a well-known problem for long-horizon prediction [1,2,3].

**References:**
1. Lambert, Nathan, Kristofer Pister, and Roberto Calandra. "Investigating compounding prediction errors in learned dynamics models." arXiv preprint arXiv:2203.09637 (2022).
2. Clavera, Ignasi, et al. "Model-based reinforcement learning via meta-policy optimization." Conference on Robot Learning. PMLR, 2018.
3. Wang, Tingwu, et al. "Benchmarking model-based reinforcement learning." arXiv preprint arXiv:1907.02057 (2019).

---

> ### Author Response · Authors · 2024-11-28
> **Summary of Rebuttal**
>
> Dear Reviewers,
>
> We would like to note that we uploaded a revision that incorporates more comparison with MTS3 (Fig. 8).   For varying SMALL time steps between 1.0 msec and 10 msec, MTS3 success rates go from 30% (@ 1.0 msec) to near 100% (@ 5 msec) and then drop down quickly to 40% at 10 msec.  For timesteps of 10 msec up to 50 msec, MTS3 success rates drop further more from 40% down to nearly 0%.  In contrast, our time-aware world model retains nearly 100% success rates for all time steps.  Even for the largest time step of 50 msec, it still retains above 90% success rates.
>
> We would like to note that we have experimented on *NINE* very different tasks of MetaWorld benchmarks (Fig. 2).  Our time-aware world model consistently outperforms baseline models.  (Ours achieves nearly 100% success rates across all time steps vs. baseline success rates drop quickly from 100% down to nearly 0%, as the time step increases from 1.0 msec up to 50 msec,(see Fig. 3, 4, 5).
>
> This is the first work that shows a novel adaptive sampling strategy (by conditioning on varying time steps in observing samples for learning) that achieves a consistently high success task learning rates (high 90% upto 100%) for World Models.  Although our motivations share some similarity with MTS3 in time scale, our algorithm in adaptive sampling on time steps offers a far more robust strategy in learning and achieves a far more consistenly high task learning rates against MTS3 (which only conditions on 2 time scales: fast and slow).   We also ran more comparison tasks (Fig. 2) and different action & observation rates and time steps, as shown in Fig. 3, 4, and 5.
>
> Robust RL is a different research with no code available.  But, based on the best known and closest work to ours, MTS3, ours achieves far higher learning rates (nearly 100%) across all time steps for varying dynamical systems and tasks.  Such a method can be incorporated into any learning architecture, including Robust RL.
>
> Lastly, we would like to note that this adaptive sampling framework conditioned on time step for observation rates is not limited or targeting solely on learning tasks in ROBOTICS, but *ALL* learning tasks involving *dynamical* systems, such as traffic prediction, simulation for rapid prototyping and design, virtual try-on, and more.   We will be happy to release our code to further research upon the accepted publication of this research.
>
> We further respectfully suggest that the reviewers please read our revision with comparison with MTS3 (as suggested), as well as comparing our manuscript with prior work like MTS3 paper for the results shown side-by-side.  We believe that the numerical comparisons on the results achieved by our TAWM are rather significant and that the learning community can all benefit tremendously from such an advance in more efficient learning of world models, as well as much higher success rates.
>
> We are very thankful to the reviewers for suggesting the comparison with MTS3 and we would be more than happy to provide additional relevant comparisons as requested.
>
> Sincerely,
>
> The Authors

---

### Meta-Review · Area_Chair_pTTX · 2024-12-22

**Metareview:**

This work proposes a way to learn a time-dependent world model with components depending on the sampling frequency. During training this is achieved by randomly varying time-steps. The same is done at test time, where the proposed method is evaluated under different time-steps and is shown to perform better than the baseline method that uses a single fixed time-step.

The consensus is that this work shows promise, but that the approach is incremental, the novelty limited, and the experiments too small to ascertain whether the proposed tweaks produce enough gains.

**Additional Comments On Reviewer Discussion:**

After the reviewing period, the reviews remained polarized. Unfortunately, the more in depth reviews did point out issues with this work. These are its incremental nature, the limited novelty and the lack of large experiments to argue that despite the two first problems the contributions are significant enough.

---

### Decision · Program_Chairs · 2025-01-22

Reject